# On the interplay between data structure and loss function in classification problems

**Stéphane d'Ascoli**
Facebook AI Research, Paris
Department of Physics, École Normale Supérieure, Paris
`stephane.dascoli@ens.fr`

**Marylou Gabrié**
New York University, New York
Flatiron Institute, New York
`mgabrie@nyu.edu`

**Levent Sagun**
Facebook AI Research, Paris

**Giulio Biroli**
Department of Physics, École Normale Supérieure, Paris

## Abstract

One of the central puzzles in modern machine learning is the ability of heavily overparametrized models to generalize well. Although the low-dimensional structure of typical datasets is key to this behavior, most theoretical studies of overparametrization focus on isotropic inputs. In this work, we instead consider an analytically tractable model of structured data, where the input covariance is built from independent blocks allowing us to tune the saliency of low-dimensional structures and their alignment with respect to the target function.

Using methods from statistical physics, we derive a precise asymptotic expression for the train and test error achieved by random feature models trained to classify such data, which is valid for any convex loss function. We study in detail how the data structure affects the double descent curve, and show that in the over-parametrized regime, its impact is greater for logistic loss than for mean-squared loss: the easier the task, the wider the gap in performance at the advantage of the logistic loss. Our insights are confirmed by numerical experiments on MNIST and CIFAR10.

## 1 Introduction

Classical wisdom teaches us that a learning model should have just the right number of parameters to learn from a dataset without overfitting it. However, recent years have seen the emergence of massively over-parametrized models which manage to generalize well on high-dimensional tasks [1, 2], somehow avoiding both the curse of dimensionality and the pitfall of overfitting. This generalization capacity in the over-parametrized regime continues to puzzle rigorous understanding, in particular for deep neural networks [3, 4, 5], despite their remarkable achievements over the past decade [6, 7, 8, 9]. Various works have given evidence of a double descent curve [5, 10, 11, 12, 13], whereby the test error first decreases as the number of parameters increases, then peaks, then decreases again monotonically. The overfitting peak occurs at the *interpolation threshold* where training error vanishes, a well-studied phenomenon in the statistical physics literature [14, 15, 16, 17, 18].

35th Conference on Neural Information Processing Systems (NeurIPS 2021).

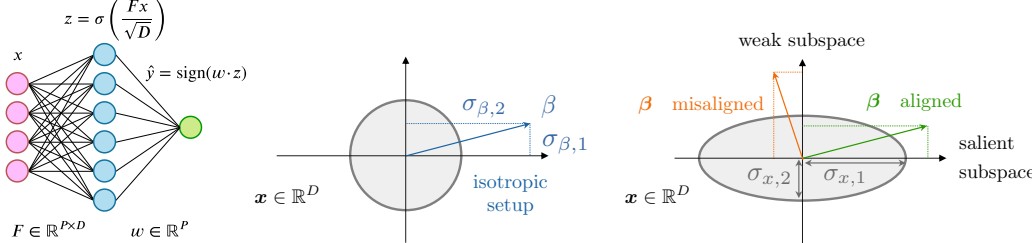

(a) Random features model          (b) Strong and weak features model of data

Figure 1: *Left:* Random feature model considered here, which can be viewed as a two-layer networks where only the second layer is trained. *Middle and Right:* Strong and weak features model considered here. Input space is decomposed into two subspaces with different variance in the anisotropic setting: a *salient* one with strong variance $\sigma_{x,1}$ and a weak one with smaller variance $\sigma_{x,2}$. The labels are given by a linear teacher $y = \text{sign}(\boldsymbol{\beta} \cdot \boldsymbol{x}/\sqrt{D})$ and flipped with a certain probability $\Delta$. We can adjust the *alignment* of data subspaces with the teacher. The task is easy when $\sigma_{\beta,1} > \sigma_{\beta,2}$, and hard in the opposite case.

Although the underlying structure of data plays a major role in the generalization ability of over-parametrized models [19], it has been little studied from a theoretical point of view. The first aspect of data structure is the distribution of the inputs: MNIST and CIFAR10 have the same number of classes and images, yet generalization is harder for CIFAR10 because the images are more complex than handwritten numbers. The second aspect is the rule between inputs and outputs: a random labelling of CIFAR10 can be learned by a neural network but offers no possibility of generalization [3]. Characterizing the structure of real-world data involves studying the interplay between these two aspects. In this direction, simple and interpretable models of structured data can prove useful to understand what underlies the behavior of generalization.

In this work, we present a simple model of learning and its analytical solution which simultaneously enables us to:

(i)   study the effect of overparametrization for a given task;

(ii)  analyze both regression and classification tasks;

(iii) control the structure of the inputs and their relationship with the labels.

To satisfy *(i)*, we need to disentangle the input dimension from the number of parameters in the learning model, which is impossible for the linear models often studied in litterature. We instead consider a random feature model [20], which was shown to exhibit double descent by [21]. To satisfy *(ii)*, we follow the lines of [22] and use an approach from statistical physics, enabling to generalize the result of [21] to any convex loss function. Finally, to satisfy *(iii)*, we introduce a block model in which the input space is subdivided into various subspaces with different variances (*saliencies*) and different correlation (*alignments*) with the labels (see Fig. 1). This model was recently studied for linear regression under the name of *strong and weak features model* [23].

Our analytical solution uses the replica method from statistical physics [24]. Albeit non-rigorous in a strict mathematical sense, this method is considered exact in theoretical physics and has been influential to statistical learning theory since the 80s, see [25] and [26] for recent reviews. Several of their conjectures have been proven exact in the last decade [27, 28, 29, 30], including in settings very close to the considered model as discussed further at the end of Section 3 [31, 32].

**Contributions**   Our main analytical contribution, presented in Section 3, is the expression of the train and test errors of a random feature model trained on data generated by the strong and weak features model. Our analysis is valid in the high-dimensional limit both for regression and classification tasks, although we focus here on the latter. We confirm that the asymptotic prediction matches simulations at finite size (see Fig. 2 and 4).

We leverage this result to present a thorough analytical study of how the data structure and the loss function interplay to shape the generalization error, and in particular how they affect the double

descent curve (Section 4). We highlight behavioral differences between square loss and logistic loss, in particular the fact that logistic loss generalizes better for easy tasks (see Fig. 3). We validate our insights via controlled experiments on the MNIST and CIFAR10 datasets described in the Section 5.

**Reproducibility**   The code to reproduce our experiments is available at `https://github.com/sdascoli/data-structure`.

## 2   Related Work

Although most theoretical studies of generalization focus on structureless data, a few exceptions exist. The strong and weak features model has recently been studied in the context of least-squares regression, both empirically [33] and theoretically [34, 35, 23]. Several intriguing observations emerge in strongly anisotropic setup: (i) several overfitting peaks can be seen  [33, 36]; (ii) the optimal ridge regularization parameter $\gamma$ can become negative [35, 37], as it becomes helpful to encourage the weights to have very different magnitudes; (iii) extra features acting as pure noise can play a beneficial role by inducing some implicit regularization [23].

The strong and weak features model also includes the setup studied in [38], called the spiked covariates model. The latter involves a small block of size $D^\eta$ ($\eta < 1$) and a large "junk" block with no correlation with the labels. The question is then: can kernel methods learn to discard the junk features, hence "beat the curse of dimensionality" in the way neural networks do? The answer was shown to depend on the strength of the junk features: when the variance of these features is small, they are not problematic, and the kernel method ignores them, effectively learning a task of effective dimensionality $D^\eta \ll D$.

More general models for the structure of data were considered in other works. In the special case of random Fourier Feature regression, [39] derived the train and test error for a general input distribution. [40] achieved a similar result in the non-parametric setting of kernel regression, which can be viewed as the limiting case of random feature regression when the number of random features $P$ goes to infinity.

The few works studying classification analytically have mostly focused on linear models, trained on linearly separable data [41, 42, 43, 44] or gaussian mixtures [45, 46]. One of the challenges in classification (in comparison to regression) is the large set of available loss functions [47, 48]. In the context of random feature models, [22] uses tools from statistical mechanics to derive the generalization loss of random features model for *any* loss function with i.i.d. Gaussian input. More recently, [32] shows that this framework could extend to more complex data distributions and learned feature maps provided that key population covariances are estimated by Monte Carlo methods. Our paper crucially builds on these contributions, by deriving a fully analytical analysis for a simple interpretable model of data structure, while also analyzing the effect of label flipping.

We also note that a few recent works investigate the roles of the loss and the structure in data in more realistic setups where theoretically robust results are harder to obtain. Several works have studied the low intrinsic dimensionality of real-world data distributions and how it impacts sample complexity for supervised tasks [49, 50, 19, 51]. The impact of the loss function is also an active research area: [52, 53] show that square loss can perform equally or better than the ubiquitous cross-entropy loss in realistic multi-class classification problems, if one rescales the weight of the correct class to emphasize its importance.

## 3   A solvable model of data structure

In this work, we focus on the random features model[1] introduced in [20]. Using tools from statistical physics, we derive the generalization error of a teacher-student task on the strong and weak features model of structured data.

---

[1]Note that this model is akin to the so-called lazy learning regime of neural networks where the weights stay close to their initial value [54]: assuming $f_{\theta_0} = 0$, we have $f_\theta(\mathbf{x}) \approx \nabla_\theta f_\theta(\mathbf{x})|_{\theta=\theta_0} \cdot (\theta - \theta_0)$ [55]. In other words, lazy learning amounts to a linear fitting problem with a random feature vector $\nabla_\theta f_\theta(\mathbf{x})|_{\theta=\theta_0}$.

**Random feature model**   The random features model can be viewed as a two-layer neural network (see Fig. 1) whose first layer is a fixed random matrix containing $P$ random feature vectors $\{\boldsymbol{F}_i \in \mathbb{R}^D\}_{i=1\dots P}$ acting on inputs $\boldsymbol{x}_\mu \in \mathbb{R}^D$:

$$\hat{y}_\mu = \sum_{i=1}^{P} w_i \sigma\left(\frac{\boldsymbol{F}_i \cdot \boldsymbol{x}_\mu}{\sqrt{D}}\right), \tag{1}$$

where $\sigma(\cdot)$ is a pointwise activation function and $w_i \in \mathbb{R}$ are the second layer weights. Elements of $\boldsymbol{F}$ are drawn i.i.d from $\mathcal{N}(0,1)$.

The second layer weights, i.e. the elements of $\boldsymbol{w} \in \mathbb{R}^P$, are trained by minimizing an $\ell_2$-regularized loss on $N$ training examples $\{\boldsymbol{x}_\mu \in \mathbb{R}^D\}_{\mu=1\dots N}$ :

$$\hat{\boldsymbol{w}} = \underset{\boldsymbol{w}}{\arg\min}\left[\epsilon_t(\boldsymbol{w}) + \frac{\lambda}{2}\|\boldsymbol{w}\|_2^2\right], \qquad \epsilon_t(\boldsymbol{w}) = \sum_{\mu=1}^{N} \ell\left(y_\mu, \hat{y}_\mu\right), \tag{2}$$

where $\ell$ denotes the logistic loss $\ell(y,\hat{y}) = \log(1+e^{-y\hat{y}})$. The target labels are given by a probabilistic teacher $y \sim \mathcal{P}_t(y|\boldsymbol{\beta} \cdot \boldsymbol{x})$ corresponding to the sign of a linear function possibly corrupted by label flipping:

$$y_\mu = \eta_\mu \,\mathrm{sign}\left(\frac{\boldsymbol{\beta} \cdot \boldsymbol{x}_\mu}{\sqrt{D}}\right), \qquad \eta_\mu = \begin{cases} 1 & \text{with probability } 1 - \Delta \\ -1 & \text{with probability } \Delta. \end{cases} \tag{3}$$

The generalization error is computed as the 0-1 loss,

$$\epsilon_g = \mathbb{E}_{\boldsymbol{x},y}\left[\mathbb{1}_{\mathrm{sign}(\hat{y}(\boldsymbol{x})),y(\boldsymbol{x})}\right]. \tag{4}$$

**Strong and weak features model**   To impose structure on the input space, we introduce a block-structured covariance matrix from which the elements of the inputs $\boldsymbol{x} \in \mathbb{R}^D$ and the teacher $\boldsymbol{\beta} \in \mathbb{R}^D$ are sampled as:

$$\boldsymbol{x} \sim \mathcal{N}(0, \Sigma_x), \qquad\qquad\qquad \boldsymbol{\beta} \sim \mathcal{N}(0, \Sigma_\beta),$$

$$\Sigma_x = \begin{bmatrix} \sigma_{x,1}\mathbb{I}_{\phi_1 D} & 0 & 0 \\ 0 & \sigma_{x,2}\mathbb{I}_{\phi_2 D} & 0 \\ 0 & 0 & \ddots \end{bmatrix}, \qquad \Sigma_\beta = \begin{bmatrix} \sigma_{\beta,1}\mathbb{I}_{\phi_1 D} & 0 & 0 \\ 0 & \sigma_{\beta,2}\mathbb{I}_{\phi_2 D} & 0 \\ 0 & 0 & \ddots \end{bmatrix}.$$

Our result presented in the rest of Section 3 is valid for an arbitrary number of blocks. In Section 4 we will focus for interpretability on the special case where we only have two blocks of sizes $\phi_1 D$ and $\phi_2 D$, with $\phi_1 + \phi_2 = 1$. We will typically be interested in the strongly anisotropic setup where the first subspace is much smaller ($\phi_1 \ll 1$), but potentially has higher *saliency* $r_x = \sigma_{x,1}/\sigma_{x,2} \gg 1$ (see Fig. 1).

**Main analytical result**   Using the replica method from statistical physics [24] and the Gaussian Equivalence Theorem (GET) [56, 21, 30, 31], we derive the generalization and training errors in the high-dimensional limit where $D, N$ and $P \to \infty$ with fixed ratios. The asymptotic generalization and training errors are given by

$$\lim_{N\to\infty} \epsilon_g = \frac{1}{\pi}\cos^{-1}\left(\frac{M}{\sqrt{\rho Q}}\right), \tag{5}$$

$$\lim_{N\to\infty} \epsilon_t = \mathbb{E}_\xi \int_{\mathbb{R}} dy \left[\mathcal{Z}^0\left(y, \xi M/\sqrt{Q}, \rho - M^2/Q\right) \ell(y, \eta(y, \sqrt{Q}\xi, V))\right] \tag{6}$$

with the proximal operator $\eta(y,a,b) = \underset{x}{\arg\min}(x-a)^2/(2b) + \ell(x,y)$, the random variable $\xi \sim \mathcal{N}(0,1)$, the functional $\mathcal{Z}^0(y,a,b) = \int_{\mathbb{R}} dx\, \mathcal{N}(x;a,b)\mathcal{P}_t(y|x)$ and the scalars

$$\rho = \sum_i \phi_i \sigma_{\beta,i}\sigma_{x,i}, \quad M = \kappa_1 \sum_i \sigma_{x,i} m_{s,i}, \quad Q = \kappa_1^2 \sum_i \sigma_{x,i} q_{s,i} + \kappa_\star^2 q_w, \quad V = \kappa_1^2 \sum_i \sigma_{x,i} v_{s,i} + \kappa_\star^2 v_w.$$

The parameters $\kappa_1, \kappa_\star$ are related to the activation function: denoting $r = \sum_i \phi_i \sigma_{x,i}$ and $\xi \sim \mathcal{N}(0, r)$, one has

$$\kappa_1 = \frac{1}{r} \, \mathbb{E}_\xi[\xi\sigma(\xi)]], \quad \kappa_\star = \sqrt{\mathbb{E}_\xi \left[\sigma(\xi)^2\right] - r\kappa_1^2}. \tag{7}$$

Besides these constants and the ones defining the data structure ($\phi_i, \sigma_{\beta,i}, \sigma_{x,i}$), the key ingredients to obtain asymptotic errors are the so-called *order parameters* $m_s, q_s, q_w, v_s$ and $v_w$. They correspond to the high-dimensional limit of the following expectations and variances (denoted by $\mathbb{V}$):

$$m_{s,i} = \lim_{D\to\infty} \frac{1}{D} \mathbb{E}_{\mathcal{P}} \left[\boldsymbol{s}_i \cdot \boldsymbol{\beta}_i\right], \quad q_{s,i} = \lim_{D\to\infty} \frac{1}{D} \mathbb{E}_{\mathcal{P}} \left[\boldsymbol{s}_i \cdot \boldsymbol{s}_i\right], \quad q_w = \lim_{P\to\infty} \frac{1}{P} \mathbb{E}_{\mathcal{P}} \left[\hat{\boldsymbol{w}} \cdot \hat{\boldsymbol{w}}\right],$$

$$v_{s,i} = \lim_{D\to\infty} \frac{1}{D} \mathbb{V}_{\mathcal{P}} \left[\boldsymbol{s}_i \cdot \boldsymbol{s}_i\right], \quad v_w = \lim_{P\to\infty} \frac{1}{P} \mathbb{V}_{\mathcal{P}} \left[\hat{\boldsymbol{w}} \cdot \hat{\boldsymbol{w}}\right],$$

where $\boldsymbol{s} = \frac{1}{\sqrt{P}} \boldsymbol{F} \hat{\boldsymbol{w}} \in \mathbb{R}^D$ and $\boldsymbol{s}_i, \boldsymbol{\beta}_i \in \mathbb{R}^{\phi_i D}$ denote the orthogonal projections of $\boldsymbol{s}$ and $\boldsymbol{\beta}$ onto subspace $i \in \{1, 2\}$ and $\mathcal{P}$ denotes the joint distribution of all random quantities in the problem (the teacher weights, the random features and the training data).

Intuitively, $\rho$ is the variance of the outputs of the teacher, $Q$ is the variance of the outputs of the student, and $M$ is their covariance. The generalization error is given by the "angle" between the teacher and the student, as expressed by Eq. 5. The order parameters allowing to obtain $Q$ and $M$ are one of the outputs of the replica computation deferred to SM B. They are obtained by solving a set of non-linear saddle-point equations (see Section B.5 of the SM). Our framework is valid for any convex loss function, although the replica equations need to be evaluated numerically in the general case. In the case of the square loss however, some simplifications arise, e.g. an explicit expression for the training error can be obtained from the order parameters (see Section B.6 of the SM).

**Steps and validity of the replica analysis** The necessary steps of the derivation are detailed in SM B. In particular, (i) we obtain an anistropic extension of the GET, (ii) conduct random matrix analysis for block matrices and finally (iii) derive the analytical saddle-point equations which yield the values of the order parameters. Our result generalizes the strategy of [22] from isotropic to anisotropic data and additionally covers the effect of label flipping. Our result is also related to [32] which establishes rigorously the replica prediction in related learning problems. A rigorous proof of our replica results is within reach: it requires a small extension of [31] (to prove the anisotropic GET derived in Section B.2 of the SM) combined with the recent results of [32]. Moreover, results in the following section show perfect agreement with numerical experiments.

## 4 Effect of data structure and loss function on double descent

In this section, we investigate how the interplay between the data structure and the loss function shapes the train and test error curves. In the main text, we only examine parameter-wise curves, where we increase the number of parameters $P$ at fixed number of data $N$. In SM. A, we also examine the sample-wise dependency by plotting the train and test error in the entire $(N, P)$ phase space.

**Modulating the teacher-data alignment** We compare three cases illustrated on Fig. 1. The first is the **isotropic** setup where $r_x = 1$ (blue curves in Fig. 2). In the two next setups, the data is anisotropic with a small subspace ($\phi_1 = 0.1$) of large variance and a large subspace ($\phi_2 = 0.9$) of small variance. The ratio of the variances $r_x = \sigma_{x,1}/\sigma_{x,2}$ is set to 10, and their values are chosen to keep the total variance of the inputs unchanged: $\frac{1}{D} \mathbb{E}[\|\boldsymbol{x}\|^2] = \phi_1\sigma_{x,1} + \phi_2\sigma_{x,2} = 1$.

We study two cases for the outputs: (i) the **aligned** scenario where the strong features are highly correlated with the labels ($r_\beta = \sigma_{\beta,1}/\sigma_{\beta,2} = 100$, green curves in Fig. 2); (ii) the **misaligned** scenario, where the strong features have low correlation with the labels ($r_\beta = 0.01$, orange curves in Fig. 2). In both cases, we choose the $\sigma_{\beta,i}$ such that the total variance of the teacher scores is unchanged: $\frac{1}{D} \mathbb{E}[(\boldsymbol{\beta} \cdot \boldsymbol{x})^2] = \phi_1\sigma_{x,1}\sigma_{\beta,1} + \phi_2\sigma_{x,2}\sigma_{\beta,2} = 1$.

**Validity at finite size** We begin by comparing our analytical predictions with the outcomes of numerical experiments, both for test loss (Fig. 2) and train loss (Fig. 4). The agreement is excellent even for moderately large dimensions $D = 100$. Note that the replica method, which relies on solving

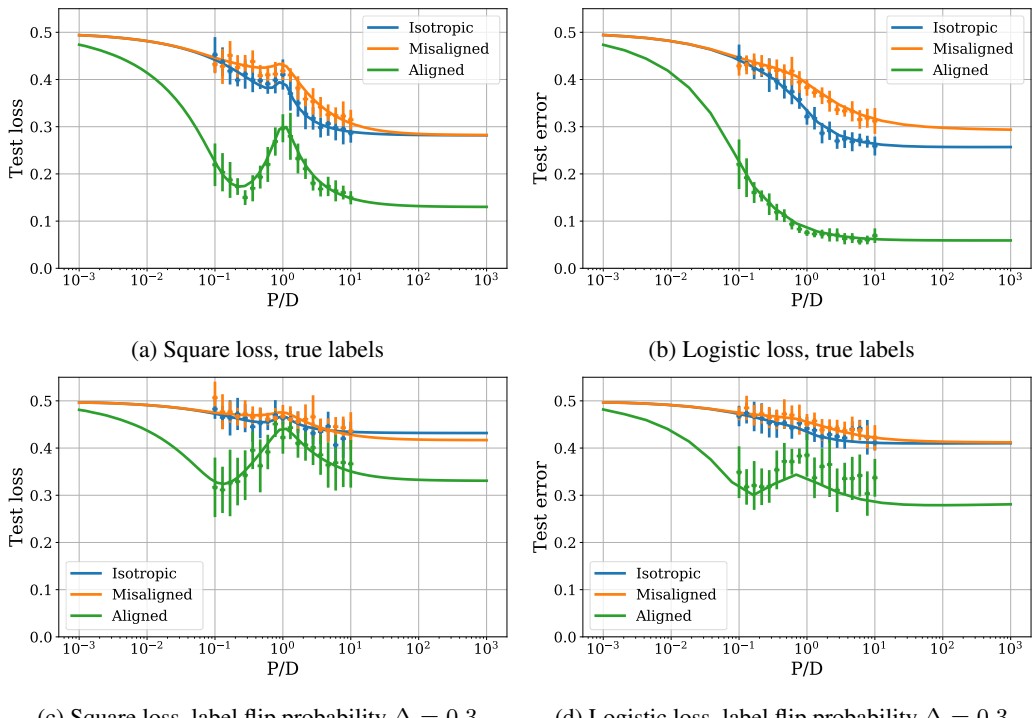

(a) Square loss, true labels

(b) Logistic loss, true labels

(c) Square loss, label flip probability $\Delta = 0.3$

(d) Logistic loss, label flip probability $\Delta = 0.3$

Figure 2: **Anisotropic data strongly affects the double descent curves.** Theoretical results (solid curves) and numerical results (dots with vertical bars denoting standard deviation over 10 runs) agree even at moderate size $D = 100$. We set $\sigma = \text{Tanh}$, $\lambda = 10^{-3}$ and $N/D = 1$.

a set of scalar fixed point equations, is also computationally efficient. It allows here to probe ratios of $P/D$ and $N/D$ far beyond what is tractable by the numerics ($P$, $D$ and $N$ only appear in the replica equations through the values of the ratios $N/P$ and $P/D$).

**Effect of data structure on generalization**    Looking at Fig. 2, a first immediate observation is that strong teacher-data alignment makes the task easier: as number of parameters $P$ increases the test loss drops earlier and eventually reaches a lower asymptotic value, both for square loss and logistic loss. In SM. A, we show that the same phenomenon occurs when varying the number of samples $N$ instead of the number of parameters $P$. These observations are in line with the results of [38] and show that relevant salient features make the problem low-dimensional with an effective dimension close to $\phi_1 D$. This setup is the most akin to real-world tasks in which the most salient features of an image are often the most relevant to its recognition. In this sense, the impressive performance of kernel methods such as the Convolutional NTK on real-world datasets [57] can be associated with the anisotropy of the data: feature learning is not indispensable to beat the curse of dimensionality if the irrelevant features are weakly salient to begin with [51].

Conversely, misalignment generally makes the task harder and increases the value of the test loss. Note however that for square loss, an interesting crossover occurs in presence of noise (panel c): the irrelevant features are detrimental from small $P$, but become helpful at large $P$, as can be seen from the orange curve reaching a lower asymptotic value than the blue curve. We associate this to the phenomenon discovered for linear regression in [23], whereby adding noisy features acts as a form of implicit regularization.

**Logistic is better than square loss**    Comparing the two loss functions, we observe two beneficial effects of using logistic loss rather than square loss.

First, we observe that the overfitting peak characteristic of the double descent curve which appears at $P = D$ for square loss is absent for logistic loss in the noiseless setting (Fig. 2), and vastly reduced in presence of noise (we use in both cases the same small amount of regularization for square and

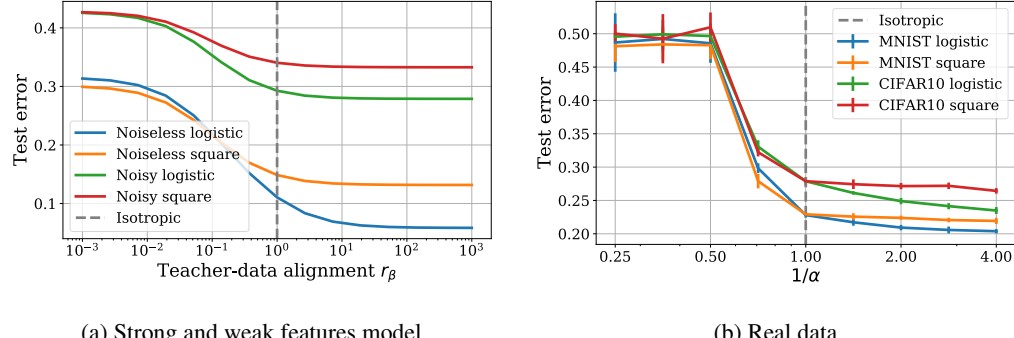

(a) Strong and weak features model  (b) Real data

Figure 3: **The easier the task, the wider the gap between logistic and square loss.** (a) Strong and weak feature model in the noiseless ($\Delta = 0$) and noisy ($\Delta = 0.3$) setups, where we make the task easier from left to right by increasing the alignment between the data and the teacher. (b) Real data (MNIST parity and CIFAR10 airplanes vs cars), where we make the task easier by decreasing the exponent $\alpha$ controlling the saliency of the top PCA components (see Sec. 5). In both cases, we considered an over-parametrized RF model ($P/D = 100$) learning from a moderate amount of data ($N/D = 1$), with $\sigma = \mathrm{Tanh}$ and $\lambda = 10^{-4}$.

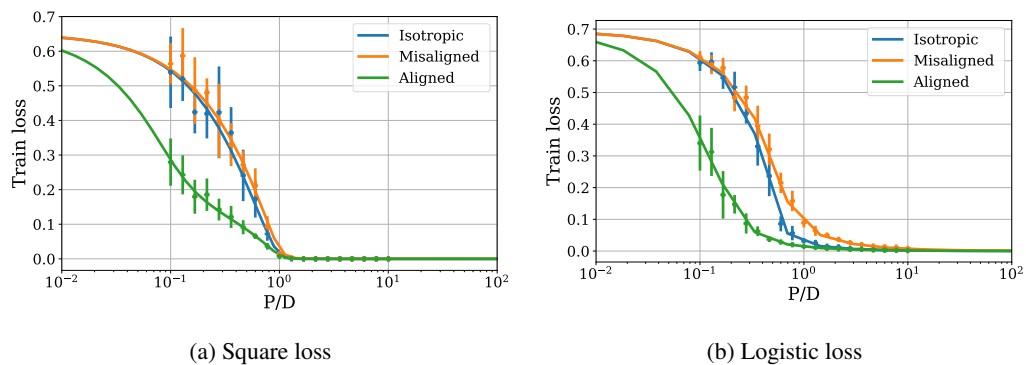

(a) Square loss  (b) Logistic loss

Figure 4: **Structure of data affects the position of the interpolation threshold for logistic loss.** We depicted the train loss curves in the noiseless setup of Fig. 2, where $\sigma = \mathrm{Tanh}$, $\lambda = 10^{-3}$, $N/D = 1$.

logistic loss). This suggests that the logistic loss exerts some form of implicit regularization, reducing the amount of overfitting.

Second, in the aligned and isotropic setups, the asymptotic test loss reached in the "kernel" regime $P/D \to \infty$ is lower with logistic loss, especially in the aligned setup. To better highlight this phenomenon, we continuously vary the teacher-data alignment in Fig. 3(a) for an overparametrized model. Logistic loss performs similarly or worse than square loss at very small alignment, but outperforms square loss as soon as the alignment is sufficient. The gap between the two then grows as we increase alignment. In other words, logistic loss is particularly powerful on tasks made easy by the structure in the data.

The better ability of logistic loss to detect structure in the data is also reflected in the train loss curves of Fig. 4. For square loss, the interpolation threshold, i.e. the point when the the train loss vanishes, occurs at $P = N$. For logistic loss, there is no interpolation threshold strictly speaking since the train loss cannot be zero. However, one can define an effective threshold as the point where the training loss reaches the near-zero plateau. Notably, this effective threshold depends on the data structure: it is lower for the aligned setup, where the data is easier to fit, and higher for the misaligned setup, where the data is harder to fit.

**Behavioral differences between losses** Further understanding of the differences between logistic and square loss can be gained thanks to the replica approach which gives access to the order parameter

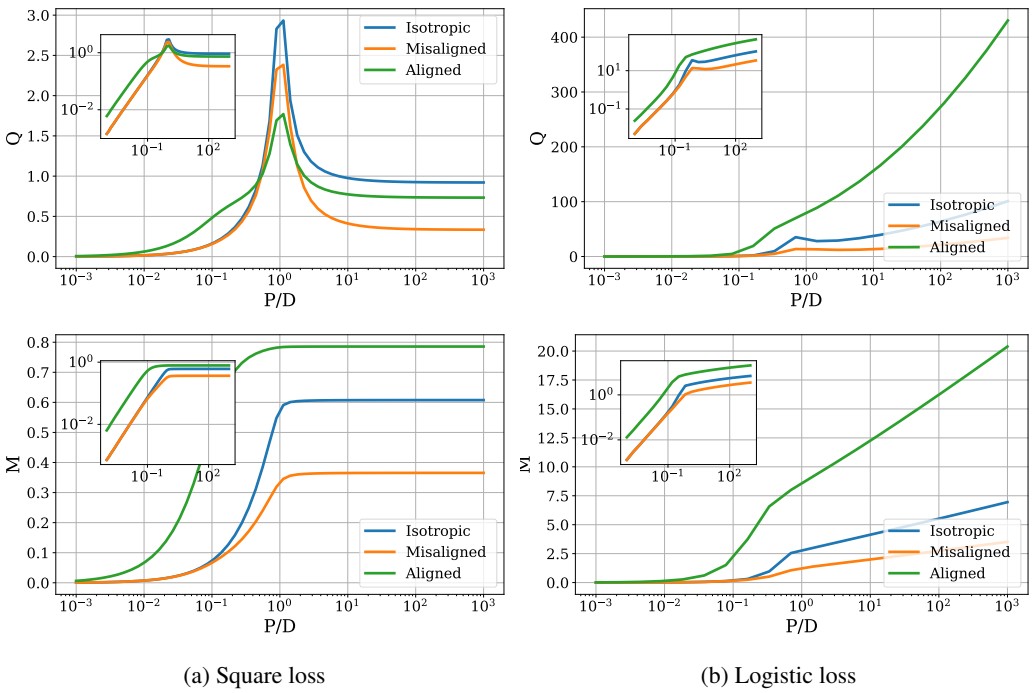

|   | (a) Square loss | (b) Logistic loss |
|---|---|---|

Figure 5: **Overparametrization causes the weights to diverge for logistic loss.** We depicted the order parameters $Q$ and $M$, quantifying the variance of the outputs of the student and their covariance with the outputs of the teacher, in the noiseless setup of Fig. 2. *Insets:* log-log plot, showing the power-law asymptotic behaviors.

$Q$ and $M$ defined in Eq. 7, see Fig. 5. For recall, $Q$ corresponds to the variance of the outputs of the student $\hat{y}$ and $M$ to their covariance with the linear scores of the teacher.

For square loss, $Q$ and $M$ increase and reach a finite value (with a peak in $Q$ at the interpolation threshold), reflecting the fact that the linearized estimator $\boldsymbol{F}\boldsymbol{w}$ converges towards a fixed norm vector more or less correlated with the teacher vector $\boldsymbol{\beta}$ depending on the depending on the data structure. For logistic loss, $M$ increases linearly with overparametrization and $Q$ increases as a power law (see logarithmic insets), reflecting the fact the the estimator endlessly grows in the direction of the teacher vector [41] as the number of parameters increases. This growth appears to shield the peak observed in $Q$ for the squared loss explaining the very mild double descent observed in Fig. 2. Interestingly, these quantities grow much faster in the aligned setup, where the estimator is more "confident" in its predictions, which also hints at the better performance of the logistic loss when the structure of data is favorable.

## 5   Numerical results

To examine the applicability of our results, we consider two realistic binary classification tasks: parity of digits in the MNIST dataset and airplanes vs cars in the CIFAR10 dataset. In both cases, we learn with an RF model in the same setup as described above. To control the alignment, we apply a PCA transformation to the inputs (keeping the top $D = 100$ components and discarding the rest), then apply the following component-wise rescaling: $\boldsymbol{x}_i \to \boldsymbol{x}_i / \operatorname{std}(\boldsymbol{x}_i)^\alpha$, where $\operatorname{std}(\boldsymbol{x}_i)$ denotes the standard deviation of feature $\boldsymbol{x}_i$ over the whole training dataset, and the exponent $\alpha$ allows us to tune the saliency of the features:

- $\alpha < 1$ yields an **aligned** scenario, since the top PCA features are naturally the most relevant;
- $\alpha = 1$ yields the **isotropic** scenario, where all features have same variance;
- $\alpha > 1$ yields a **misaligned** scenario, since the strong features will become weak and the weak features become strong.

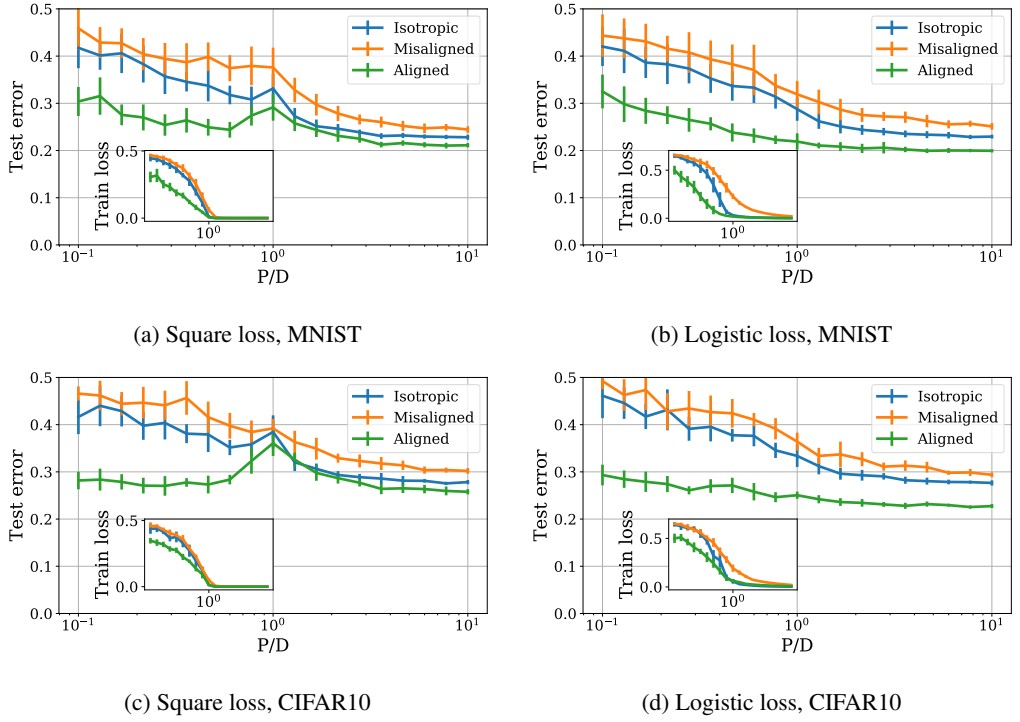

(a) Square loss, MNIST

(b) Logistic loss, MNIST

(c) Square loss, CIFAR10

(d) Logistic loss, CIFAR10

Figure 6: **Test error and train loss (inset) on realistic tasks.** *Top:* MNIST dataset with labels given by the digit parity. *Bottom:* CIFAR10 dataset with airplanes and cars. We synthetically reproduce the isotropic, aligned and misaligned scenarios by applying a PCA transformation to the inputs and tuning how salient the largest PCA features are compared to the smallest PCA features (see Sec. 5). We set $\sigma = \mathrm{Tanh}$, $\lambda = 10^{-3}$ and $N/D = 1$.

The corresponding train and test error curves are shown in Fig. 6 (we set $\alpha = 0$ for the aligned scenario and $\alpha = 1.5$ for the misaligned scenario). Remarkably, we recover many of the phenomenological features described previously. The test error drops earlier and reaches a lower asymptotic value in the aligned setup, and conversely reaches a higher value in the misaligned setup. We observe a double descent curve for square loss, but the peak is suppressed for logistic loss. The location of the interpolation threshold depends on the teacher-data alignment for logistic loss, whereas it does not for square loss. Finally, logistic loss has a lower asymptotic error than square loss in the aligned setup, signalling that it is favorable for "easy" data distributions.

To strengthen the latter observation, we vary continuously the difficulty of the task by adjusting the exponent $\alpha$ and show the results in Fig. 3(b) (increasing $\alpha$ makes the task harder). As observed analytically in Fig. 3(a), the gap between square loss and logistic loss increases as we decrease $\alpha$.

# 6 Conclusion

In this work, we studied how the loss function interplays with the data structure to shape the generalization curve of random feature models. Our results show strong behavioral differences between quadratic and logistic loss, the latter performing particularly well for easy datasets where most of the information comes from low-dimension projections of the inputs.

As a possible direction of future work, we conclude with the observation that our results, which apply to random feature (or lazy learning) tasks, appear in contrast with those of [52], which suggest that cross-entropy losses can be traded at no cost for quadratic losses in modern deep learning tasks, which are known to have low intrinsic dimensionality [19]. This opens up an interesting direction for future work: does feature learning help quadratic losses by better capturing the low-dimensional structure of the inputs, as suggested by [51]? Or does the key difference reside in the multi-class nature of practical classification problems [53]?

**Acknowledgements** We thank Berfin Simsek, Armand Joulin, Bruno Loureiro and Federica Gerace for helpful discussions. SD and GB acknowledge funding from the French government under management of Agence Nationale de la Recherche as part of the "Investissements d'avenir" program, reference ANR-19-P3IA-0001 (PRAIRIE 3IA Institute). MG acknowledges funding from the Flatiron Institute. We also thank Lenka Zdeborova and Florent Krzakala, and Les Houches Summer School for organizing a workshop on Statistical Physics and Machine Learning where this work was partially done.

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
