[58] Stéphane d'Ascoli, Levent Sagun, and Giulio Biroli. Triple descent and the two kinds of overfitting: where and why do they appear? In *Advances in Neural Information Processing Systems*, volume 33, pages 3058–3069. Curran Associates, Inc., 2020.

[59] Sebastian Goldt, Marc Mézard, Florent Krzakala, and Lenka Zdeborová. Modelling the influence of data structure on learning in neural networks. *arXiv preprint arXiv:1909.11500*, 2019.

## A   Phase spaces

**Comparing square and logistic**   In Figs. 7 and 8, we show how the various observables of interest evolve in the $(N, P)$ phase space, respectively from square loss and logistic loss. To show that our results generalize to different activation functions and regularization levels, we choose $\sigma = \mathrm{ReLU}$ and $\lambda = 0.1$. We make the following observations:

1. **Test error**: the phase-space is almost symmetric in the isotropic setup, with the double descent peak clearly visible at $N = P$ (dashed grey line) for square loss but strongly attenuated for logistic loss. Interestingly, for logistic loss, the peak appears at $N > P$ in the noiseless setup as observed in [22], but shifts to $N = P$ in presence of noise. In the anisotropic setup, the phase space dissymetrizes, with a wide overfitting region emerging in the overparametrized regime around $N = D$ (solid grey line), as observed in [58] for isotropic regression tasks. This overfitting is strongly regularized for logistic loss, explaining its superiority on structured datasets

2. **Train loss**: Here a strong difference appears between square loss and logistic loss. For square loss, the overparametrized region $P > N > D$ reaches zero training loss, and the interpolation threshold clearly appears at $P = N$. For logistic loss, the interpolation threshold depends more strongly on the data structure: it shifts up when we increase the label noise and shifts down when we increase the anisotropy.

3. **Order parameter** $Q$: recall that this observable represents the variance of the student's outputs. Here the phase space looks completely different for the two loss functions. For square loss, the phase space is almost symmetric in the noiseless isotropic setup and dissymetrizes in presence of noise or anisotropy, with a peak appearing at $N = D$, forming the "linear peak" in the test error [58]. For logistic loss, the phase space is not symmetric: $Q$ becomes very large in the overparametrized regime, leading to overconfidence, as observed in Fig. 5 of the main text.

4. **Order parameter** $M$: recall that this observable represents the covariance between the student's outputs and the teacher's outputs, i.e. the dot product of their corresponding vectors. Again, behavior is very different for square loss and logistic loss. For square loss, the covariance increases symmetrically when increasing $P$ or $N$, reaching its maximal value respectively at $P = D$ and $N = D$ in the isotropic setup, or earlier in the aligned setup. For logistic loss, the phase space is more complex due to the fact that the norm of the student diverges in the overparametrized regime.

**Varying the anisotropy**   Fig. 9 illustrates the modification of the phase space of the random features model trained on the strong and weak features model as the saliency $r_x$ of the $\phi_1 D = 0.01D$ relevant features ($r_\beta = 1000$) is gradually increased. At $r_x = 1$ (panel (a)), the data is isotropic and the phase space is symmetric. When $r_x \to \infty$, all the variance goes in the salient subspace and we are left with an isotropic task of dimensionality $\phi_1 D$: the phase space is symmetric again (panel (d)). In between these two extreme scenarios, we see the asymmetry appear in the phase space under the form of an overfitting peak at $N = D$, as the irrelevant features come into play (panels (b) and (c)).

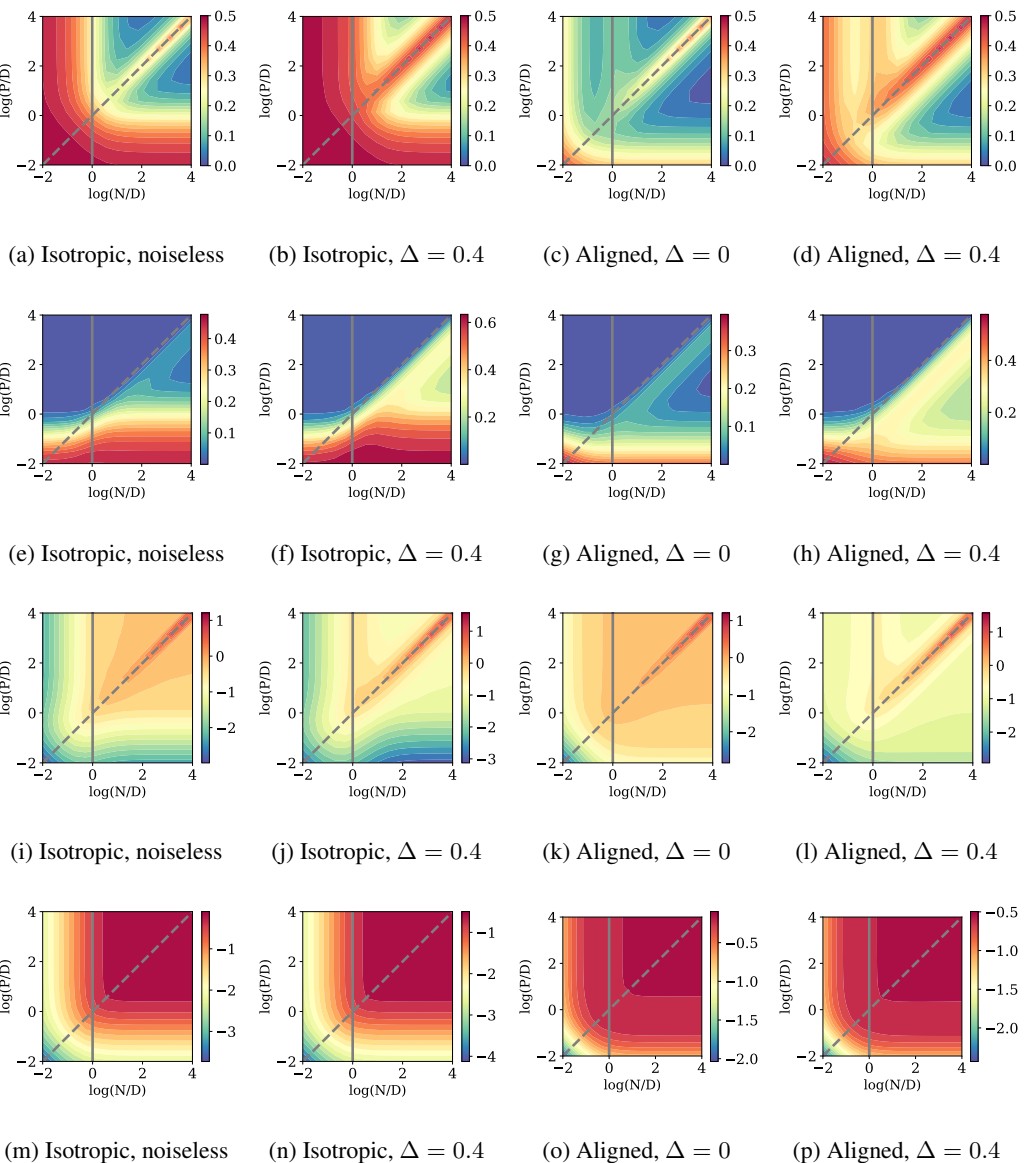

Figure 7: **Square loss phase spaces.** We studied the classification task for $\sigma = \mathrm{ReLU}$, $\lambda = 0.1$. In the anisotropic phase spaces we set $\phi_1 = 0.01$, $r_x = 100$, $r_\beta = 100$. *First row:* test error. *Second row:* train error. *Third row:* Q. *Fourth row:* M. The solid and dashed grey lines represent the $N = D$ and $N = P$ lines, where one can find overfitting peaks [58]. For $Q$ and $M$, the colormaps are logarithmic.

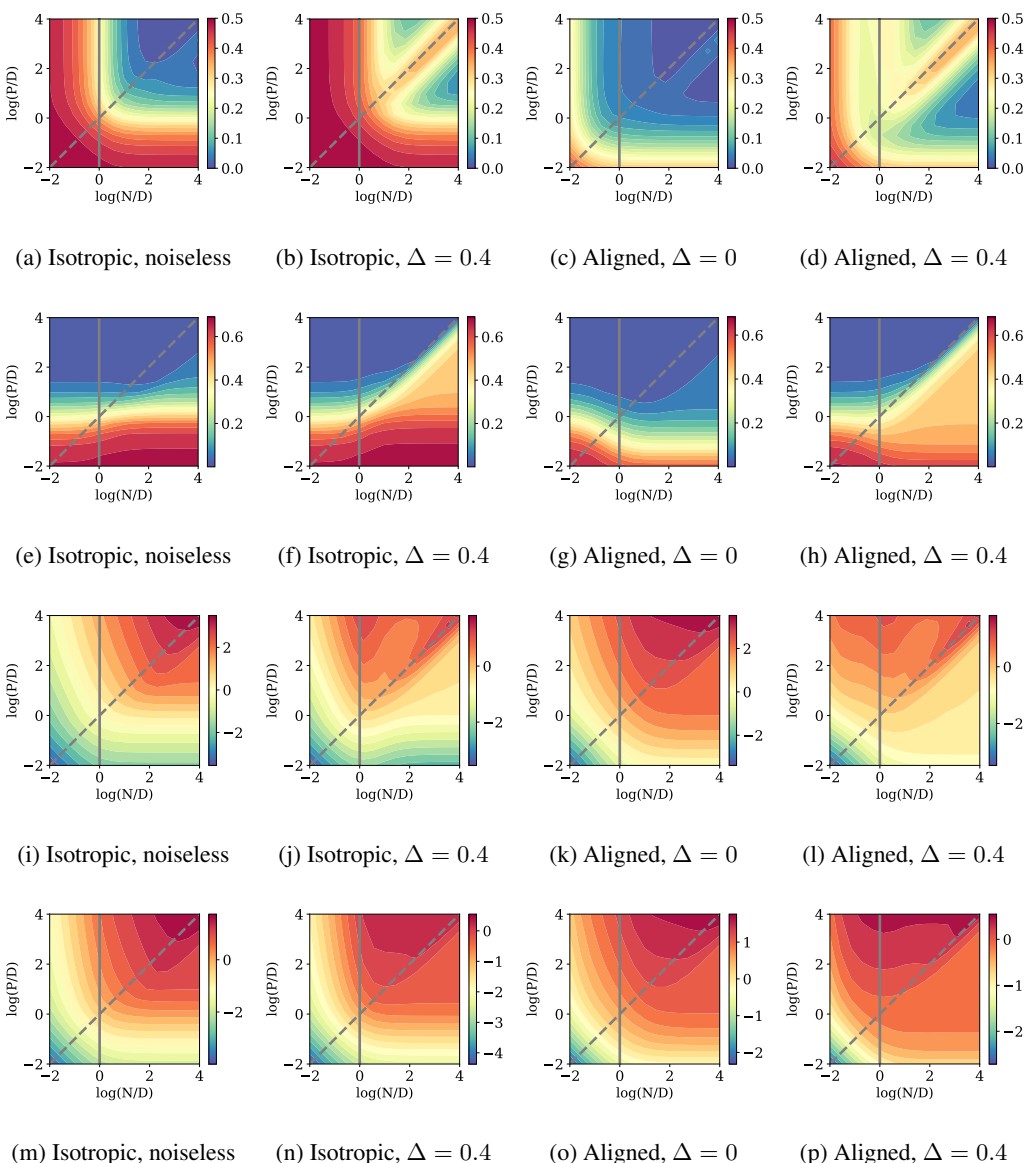

(a) Isotropic, noiseless     (b) Isotropic, $\Delta = 0.4$     (c) Aligned, $\Delta = 0$     (d) Aligned, $\Delta = 0.4$

(e) Isotropic, noiseless     (f) Isotropic, $\Delta = 0.4$     (g) Aligned, $\Delta = 0$     (h) Aligned, $\Delta = 0.4$

(i) Isotropic, noiseless     (j) Isotropic, $\Delta = 0.4$     (k) Aligned, $\Delta = 0$     (l) Aligned, $\Delta = 0.4$

(m) Isotropic, noiseless     (n) Isotropic, $\Delta = 0.4$     (o) Aligned, $\Delta = 0$     (p) Aligned, $\Delta = 0.4$

Figure 8: **Logistic loss phase spaces.** We studied the classification task for $\sigma = \mathrm{ReLU}$, $\lambda = 0.1$. In the Aligned phase spaces we set $\phi_1 = 0.01$, $r_x = 100$, $r_\beta = 100$. *First row:* test error. *Second row:* train error. *Third row:* Q. *Fourth row:* M. The solid and dashed grey lines represent the $N = D$ and $N = P$ lines, where one can find overfitting peaks [58]. For $Q$ and $M$, the colormaps are logarithmic.

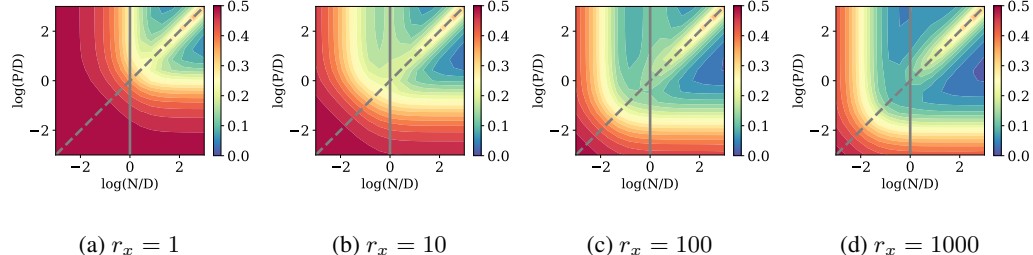

(a) $r_x = 1$  (b) $r_x = 10$  (c) $r_x = 100$  (d) $r_x = 1000$

Figure 9: Increasing the saliency of the first subspace from 1 to $\infty$, the asymmetry forms then vanishes as the data becomes effectively isotropic in smaller dimension. We study the classification task for $\sigma = \text{ReLU}$, $\lambda = 0.1$, $\phi_1 = 0.01$, $r_\beta = 1000$ and $\Delta = 0$.

# B    Analytical derivations

The following sections present the analytical derivations of this work. First, we provide an outline of the computation of the generalization error. Second, we describe our extension of the Gaussian Equivalence Theorem to anisotropic data, a key ingredient to handle the strong and weak features model. The three following sections develop the steps of the replica computation, from the Gibbs formulation of the learning objective, to the set of self-consistent equations to obtain the order parameters. Lastly, we explain how to also obtain the asymptotic training loss from the output of the replica computation.

## B.1    Outline

In the main text, we study the strong and weak features scenario with two blocks, and only for classification tasks, but our derivation will be performed here in full generality.

**Setup and notations**    We recall notations: $D$ is the input dimension, $P$ is the number of random features and $N$ is the number of training examples. We consider the high-dimensional limit where $D, P, N \to \infty$ and $\gamma = \frac{D}{P}, \alpha = \frac{N}{P}$ are of order one. The inputs $\boldsymbol{x} \in \mathbb{R}^D$ and the elements of the teacher vectors $\boldsymbol{\beta} \in \mathbb{R}^D$ are sampled from a block-structured covariance matrix :

$$\boldsymbol{x} \sim \mathcal{N}(0, \Sigma_x), \quad \Sigma_x = \begin{bmatrix} \sigma_{x,1}\mathbb{I}_{\phi_1 D} & 0 & 0 \\ 0 & \sigma_{x,2}\mathbb{I}_{\phi_2 D} & 0 \\ 0 & 0 & \ddots \end{bmatrix} \tag{8}$$

$$\boldsymbol{\beta} \sim \mathcal{N}(0, \Sigma_\beta), \quad \Sigma_\beta = \begin{bmatrix} \sigma_{\beta,1}\mathbb{I}_{\phi_1 D} & 0 & 0 \\ 0 & \sigma_{\beta,2}\mathbb{I}_{\phi_2 D} & 0 \\ 0 & 0 & \ddots \end{bmatrix} \tag{9}$$

$$\tag{10}$$

The output of the random feature model is given by:

$$\hat{y}_\mu = \hat{f}\left(\sum_{i=1}^{P} \boldsymbol{w}_i \sigma\left(\frac{\boldsymbol{F}_i \cdot \boldsymbol{x}_\mu}{\sqrt{D}}\right)\right) \equiv \hat{f}\left(\boldsymbol{w} \cdot \boldsymbol{z}_\mu\right), \tag{11}$$

where $\sigma(\cdot)$ is a pointwise activation function, $\hat{f}$ is an output function and we introduced the notation $\boldsymbol{z}_\mu \in \mathbb{R}^P$. Elements of $\boldsymbol{F}$ are drawn i.i.d from $\mathcal{N}(0, 1)$. The labels are given by a linear ground truth, modulated by an output function $f^0$ and possibly corrupted by noise through a probabilistic channel $\mathcal{P}$:

$$y_\mu \sim \mathcal{P}\left(\cdot \left| f^0\left(\frac{\boldsymbol{\beta} \cdot \boldsymbol{x}_\mu}{\sqrt{D}}\right)\right.\right). \tag{12}$$

The second layer weights, i.e. the elements of $\boldsymbol{w} \in \mathbb{R}^P$, are trained by minimizing an $\ell_2$-regularized loss on $N$ training examples $\{\boldsymbol{x}_\mu \in \mathbb{R}^D\}_{\mu=1...N}$ :

$$\hat{\boldsymbol{w}} = \underset{\boldsymbol{w}}{\mathrm{argmin}}\left[\epsilon_t(\boldsymbol{w})\right], \quad \epsilon_t(\boldsymbol{w}) = \sum_{\mu=1}^{N} \ell\left(y^\mu, \boldsymbol{w} \cdot \boldsymbol{z}_\mu\right) + \frac{\lambda}{2}\|\boldsymbol{w}\|_2^2. \tag{13}$$

We consider both regression tasks, where $\hat{f} = f^0 = \mathrm{id}$, and binary classification tasks, $\hat{f} = f^0 = \mathrm{sign}$. In the first case, $\ell$ is the square loss and the noise is additive. In the second case, $\ell$ can be any type of loss (square, hinge, logistic) and the noise amounts to random label flipping.

The generalization error is given by

$$\epsilon_g = \frac{1}{2^k} \mathbb{E}_{\boldsymbol{x},y}\left[\left(\hat{y}\left(\boldsymbol{x}\right) - y\right)^2\right] \tag{14}$$

with $k = 1$ for regression tasks (mean-square error) and $k = 2$ for binary classification tasks (zero-one error).

In the following, we denote as $\boldsymbol{x}|_i, \boldsymbol{\beta}|_i$ the orthogonal projection of $\boldsymbol{x}$ and $\boldsymbol{\beta}$ onto the subspace $i$ of $\mathbb{R}^D$. For example, $\boldsymbol{x}|_1$ amounts to the first $\phi_1 D$ components of $\boldsymbol{x}$.

**Steps of the derivation of the generalization error**    The key observation is that the test error can be rewritten, in the high-dimensional limit, in terms of so-called *order parameters*:

$$\epsilon_g = \frac{1}{2^k} \mathbb{E}_{\boldsymbol{x}^{\text{new}}, y^{\text{new}}} \left( y^{\text{new}} - \hat{f} \left( \frac{1}{\sqrt{P}} \sigma \left( \frac{\mathrm{F}^\top \boldsymbol{x}^{\text{new}}}{\sqrt{D}} \right) \cdot \hat{\boldsymbol{w}} \right) \right)^2 \tag{15}$$

$$= \frac{1}{2^k} \int \mathrm{d}\nu \int \mathrm{d}\lambda P(\nu, \lambda)(f^0(\nu) - \hat{f}(\lambda))^2 \tag{16}$$

where we defined

$$\lambda = \frac{1}{\sqrt{P}} \sigma \left( \frac{\mathrm{F}^\top \boldsymbol{x}^{\text{new}}}{\sqrt{D}} \right) \cdot \hat{\boldsymbol{w}} \in \mathbb{R}, \quad \nu = \frac{1}{\sqrt{D}} \boldsymbol{x}^{\text{new}} \cdot \boldsymbol{\beta} \in \mathbb{R} \tag{17}$$

and the joint probability distribution

$$P(\nu, \lambda) = \mathbb{E}_{\boldsymbol{x}^{\text{new}}} \left[ \delta \left( \nu - \frac{1}{\sqrt{D}} \boldsymbol{x}^{\text{new}} \cdot \boldsymbol{\beta} \right) \delta \left( \lambda - \frac{1}{\sqrt{P}} \sigma \left( \frac{\mathrm{F}^\top \boldsymbol{x}^{\text{new}}}{\sqrt{D}} \right) \cdot \hat{\boldsymbol{w}} \right) \right]. \tag{18}$$

The key objective to calculate the test error is therefore to obtain the joint distribution $P(\nu, \lambda)$. To do so, our derivation is organized as follows.

1. In Sec. B.2, we adapt the Gaussian equivalence theorem [56, 21, 59, 31] to the case of anisotropic data. The latter shows that $P(\nu, \lambda)$ is a joint gaussian distribution whose covariance depends only on the following *order parameters*:

$$m_{s,i} = \frac{1}{D} \boldsymbol{s}|_i \cdot \boldsymbol{\beta}|_i, \quad q_{s,i} = \frac{1}{D} \boldsymbol{s}|_i \cdot \boldsymbol{s}|_i, \quad q_w = \frac{1}{P} \hat{\boldsymbol{w}} \cdot \hat{\boldsymbol{w}},$$

where we denoted $\boldsymbol{s} = \frac{1}{\sqrt{P}} \boldsymbol{F} \hat{\boldsymbol{w}} \in \mathbb{R}^D$ and the index $i$ refers to the subspace of $\mathbb{R}^D$ the vectors are projected onto.

2. In Sec. B.3, we recast the optimization problem as a Gibbs measure over the weights, from which one can sample the average value of the order parameters $m_s, q_s, q_w$. Thanks to the convexity of the problem, this measure concentrates around the solution of the optimization problem in the limit of high temperature, see [22].

3. In Sec. B.4, we leverage tools from random matrix theory to derive the saddle-point equations allowing the obtain the values of the order parameters

4. In Sec. B.5, we give the explicit expressions of the saddle-point equations for the two cases studied in the main text: square loss regression and classification

5. In Sec. B.6, we also show how to obtain the train error from the order parameters.

Once the order parameters are known, the joint law of $\lambda, \nu$ is determined as explained in the main text:

$$P(\lambda, \nu) = \mathcal{N}(0, \Sigma), \quad \Sigma = \begin{pmatrix} \rho & M \\ M & Q \end{pmatrix}. \tag{19}$$

It is then easy to perform the Gaussian integral giving the generalization error.

For the regression problem where $\hat{f} = f^0 = \text{id}$, we have:

$$\epsilon_g = \frac{1}{2} (\rho + Q - 2M)$$

For the classification problem where $\hat{f} = f^0 = \text{sgn}$, we have:

$$\epsilon_g = \frac{1}{\pi} \cos^{-1} \left( \frac{M}{\sqrt{\rho Q}} \right)$$

.

## B.2 The anisotropic Gaussian Equivalence Theorem

We now present the derivation of the anisotropic Gaussian Equivalence Theorem. This is a key ingredient for the replica analysis presented in the next section.

Define, for a Gaussian input vector $\boldsymbol{x} \in \mathbb{R}^D$, the family of vectors indexed by $a = 1 \ldots r$:

$$\lambda^a = \frac{1}{\sqrt{P}} \boldsymbol{w}^a \cdot \sigma\left(\frac{\boldsymbol{F}\boldsymbol{x}}{\sqrt{D}}\right) \in \mathbb{R}, \quad \nu = \frac{1}{\sqrt{D}} \boldsymbol{x} \cdot \boldsymbol{\beta} \in \mathbb{R} \tag{20}$$

Using the Gaussian equivalence principle, we can obtain expectancies for this family in the high-dimensional limit:

$$(\nu, \lambda^a) \sim \mathcal{N}(0, \Sigma), \quad \Sigma^{ab} = \begin{pmatrix} \rho & M^a \\ M^a & Q^{ab} \end{pmatrix} \in \mathbb{R}^{(r+1)\times(r+1)} \tag{21}$$

$$\rho = \sum_i \phi_i \beta|_i \sigma_{x,i}, \quad M^a = \kappa_1 \sum_i \sigma_{x,i} m^a_{s,i}, \quad Q^{ab} = \kappa_1^2 \sum_i \sigma_{x,i} q^{ab}_{s,i} + \kappa_\star^2 q^{ab}_w \tag{22}$$

$$m^a_{s,i} = \frac{1}{D} \boldsymbol{s}^a|_i \cdot \boldsymbol{\beta}|_i, \quad q^{ab}_{s,i} = \frac{1}{D} \boldsymbol{s}^a|_i \cdot \boldsymbol{s}^b|_i, \quad q^{ab}_w = \frac{1}{P} \boldsymbol{w}^a \cdot \boldsymbol{w}^b \tag{23}$$

where

$$\kappa_0 = \mathop{\mathbb{E}}_{z\sim\mathcal{N}(0,r)}[\sigma(z)], \quad \kappa_1 = \frac{1}{r} \mathop{\mathbb{E}}_{z\sim\mathcal{N}(0,r)}[z\sigma(z)]], \quad \kappa_\star = \sqrt{\mathop{\mathbb{E}}_{z\sim\mathcal{N}(0,\sum_i \sigma_{x,i}\phi_i)}[\sigma(z)^2] - \kappa_0^2 - r\kappa_1^2} \tag{24}$$

$$r = \sum_i \phi_i \sigma_{x,i}, \quad \boldsymbol{s}^a = \frac{1}{\sqrt{P}} \boldsymbol{F}\boldsymbol{w}^a \in \mathbb{R}^D, \quad \boldsymbol{s}^a_i = P_{E_i}\boldsymbol{s}^a, \quad \boldsymbol{\beta}|_i = P_{E_i}\boldsymbol{\beta} \tag{25}$$

$$\tag{26}$$

**Isotropic setup** Let us start in the setup where we only have one block, of unit variance.

By rotational invariance, we can write :

$$\mathbb{E}_{\boldsymbol{x}}\left[\sigma\left(\frac{\boldsymbol{x}\cdot F_\mu}{\sqrt{D}}\right)\sigma\left(\frac{\boldsymbol{x}\cdot F_\nu}{\sqrt{D}}\right)\right] = f\left(\frac{F_\mu \cdot F_\nu}{D}\right) \equiv f(q_{\mu\nu}) \tag{27}$$

$$= \delta_{\mu\nu}f(1) + (1 - \delta_{\mu\nu})(f(0) + f'(0)q_{\mu\nu}) + o(q_{\mu\nu}) \tag{28}$$

$$= f(0) + f'(0)q_{\mu\nu} + \delta_{\mu\nu}(f(1) - f(0) - f'(0)) \tag{29}$$

Therefore, the expectancy is the same as the "Gaussian equivalent model" where we replace

$$\sigma\left(\frac{\boldsymbol{x}\cdot F_\mu}{\sqrt{D}}\right) \rightarrow \sqrt{f(0)} + \sqrt{f'(0)}\left(\frac{\boldsymbol{x}\cdot F_\mu}{\sqrt{D}}\right) + \sqrt{(f(1) - f(0) - f'(0)}\xi_\mu, \quad \xi_\mu \sim \mathcal{N}(0,1) \tag{30}$$

What remains is to find the expression of $f(0)$, $f(1)$ and $f'(0)$. If $q_{\mu\nu} = 1$, then $F_\mu$ and $F_\nu$ are the same vector and clearly $f(1) = \mathbb{E}_z\left[\sigma(z)^2\right]$.

Otherwise, we use the fact that $x_\mu = \frac{\boldsymbol{x}\cdot F_\mu}{\sqrt{D}}$ and $x_\nu = \frac{\boldsymbol{x}\cdot F_\nu}{\sqrt{D}}$ are correlated gaussian variables:

$$\mathbb{E}\left[x_\mu^2\right] = 1, \quad \mathbb{E}\left[x_\mu x_\nu\right] = \frac{F_\mu \cdot F_\nu}{D} \equiv q_{\mu\nu} \sim O\left(\frac{1}{\sqrt{D}}\right) \tag{31}$$

Therefore we can parametrize as follows,

$$x_\mu = \eta_\mu \sqrt{r - q_{\mu\nu}} + z\sqrt{q_{\mu\nu}} \sim \eta_\mu(1 - \frac{1}{2}q_{\mu\nu}) + z\sqrt{q_{\mu\nu}} \tag{32}$$

$$x_\nu = \eta_\nu \sqrt{r - q_{\mu\nu}} + z\sqrt{q_{\mu\nu}} \sim \eta_\mu(1 - \frac{1}{2}q_{\mu\nu}) + z\sqrt{q_{\mu\nu}} \tag{33}$$

$$\tag{34}$$

where $\eta_\mu, \eta_\nu, z \sim \mathcal{N}(0,1)$. To calculate $f(0), f'(0)$, we can expand the nonlinearity,

$$f(q_{\mu\nu}) \equiv f(0) + f'(0)q_{\mu\nu} + o(q_{\mu\nu}^2) \tag{35}$$

$$= \mathbb{E}\left[\left(\sigma(\eta_\mu) + \sigma'(\eta_\mu)\left(-\frac{q}{2}\eta_\mu + z\sqrt{q}_{\mu\nu}\right) + \frac{\sigma''(\eta_\mu)}{2}\left(z^2 q_{\mu\nu}\right)\right)\right. \tag{36}$$

$$\left.\left(\sigma(\eta_\nu) + \sigma'(\eta_\nu)\left(-\frac{q}{2}\eta_\nu + z\sqrt{q}_{\mu\nu}\right) + \frac{\sigma''(\eta_\nu)}{2}\left(z^2 q_{\mu\nu}\right)\right)\right] \tag{37}$$

$$= \mathbb{E}_\eta\left[\sigma(\eta)\right]^2 + q_{\mu\nu}\left(\mathbb{E}[\sigma'(\eta)]^2 + \mathbb{E}[\sigma''(\eta)]\mathbb{E}[\sigma(\eta)] - \mathbb{E}[\sigma'(\eta)\eta]\mathbb{E}[\sigma(\eta)]\right) + o(q_{\mu\nu}^2) \tag{38}$$

$$\tag{39}$$

But since $\mathbb{E}[\sigma(\eta)\eta] = \mathbb{E}[\sigma'(\eta)]$, the two last terms cancel out and we are left with

$$f(0) = \mathbb{E}[\sigma(z)]^2, \qquad f'(0) = \mathbb{E}[\sigma(z)\eta]^2 \tag{40}$$

$$\mathbb{E}_{\boldsymbol{x}}\left[\sigma\left(\frac{\boldsymbol{x}\cdot F_\mu}{\sqrt{D}}\right)\sigma\left(\frac{\boldsymbol{x}\cdot F_\nu}{\sqrt{D}}\right)\right] = \kappa_0^2 + \kappa_1^2\frac{F_\mu\cdot F_\nu}{D} + \kappa_\star^2\delta_{\mu\nu} \tag{41}$$

**Anisotropic setup**   Now in the block setup,

$$\mathbb{E}\left[x_\mu^2\right] = \sum_i \sigma_{x,i}\phi_i \equiv r, \qquad \mathbb{E}\left[x_\mu x_\nu\right] = \sum_i \sigma_{x,i}\frac{F_\mu^i\cdot F_\nu^i}{D} = \sum_i \sigma_{x,i}q_{\mu\nu}^i \equiv q_{\mu\nu} \sim O\left(\frac{1}{\sqrt{D}}\right) \tag{42}$$

Therefore we can parametrize as follows,

$$x_\mu = \eta_\mu\sqrt{1 - \frac{q_{\mu\nu}}{r}} + z\sqrt{\frac{q_{\mu\nu}}{r}} \sim \eta_\mu(1 - \frac{1}{2}\frac{q_{\mu\nu}}{r}) + z\sqrt{\frac{q_{\mu\nu}}{r}} \tag{43}$$

$$x_\nu = \eta_\nu\sqrt{1 - \frac{q_{\mu\nu}}{r}} + z\sqrt{\frac{q_{\mu\nu}}{r}} \sim \eta_\mu(1 - \frac{1}{2}\frac{q_{\mu\nu}}{r}) + z\sqrt{\frac{q_{\mu\nu}}{r}} \tag{44}$$

$$\tag{45}$$

where $\eta_\mu, \eta_\nu, z \sim \mathcal{N}(0,r)$. As before,

$$f(q_{\mu\nu}) = \mathbb{E}_\eta\left[\sigma(\eta)\right]^2 + \frac{q_{\mu\nu}}{r}\left(\mathbb{E}[\sigma'(\eta)]^2\,\mathbb{E}[z^2] + \mathbb{E}[\sigma''(\eta)]\mathbb{E}[\sigma(\eta)]\,\mathbb{E}[z^2] - \mathbb{E}[\sigma'(\eta)\eta]\mathbb{E}[\sigma(\eta)]\right) + o(q_{\mu\nu}^2) \tag{46}$$

$$\tag{47}$$

Now since $\mathbb{E}[z^2] = r$ and $\mathbb{E}[\sigma(\eta)\eta] = r\mathbb{E}[\sigma'(\eta)]$, the two last terms do not cancel out like before and we have :

$$f(0) = \mathbb{E}[\sigma(z)]^2, \qquad f'(0) = \frac{1}{r}\mathbb{E}[\sigma(z)\eta]^2\mathbb{E}[z^2] = \frac{1}{r^2}\mathbb{E}[\sigma(z)z]^2 \tag{48}$$

Finally we have

$$\mathbb{E}_{\boldsymbol{x}}\left[\sigma\left(\frac{\boldsymbol{x}\cdot F_\mu}{\sqrt{D}}\right)\sigma\left(\frac{\boldsymbol{x}\cdot F_\nu}{\sqrt{D}}\right)\right] = \kappa_0^2 + \kappa_1^2\sum_i \sigma_{x,i}\frac{F_\mu^i\cdot F_\nu^i}{D} + \kappa_\star^2\delta_{\mu\nu} \tag{49}$$

$$\kappa_0 = \mathbb{E}_{z\sim\mathcal{N}(0,r)}[\sigma(z)], \quad \kappa_1 = \frac{1}{r}\mathbb{E}_{z\sim\mathcal{N}(0,r)}[z\sigma(z)]], \quad \kappa_\star = \sqrt{\mathbb{E}_{z\sim\mathcal{N}(0,r)}[\sigma(z)^2] - \kappa_0^2 - r\kappa_1^2} \tag{50}$$

$$\tag{51}$$

## B.3 Gibbs formulation of the problem

To obtain the test error, we need to find the typical value of the order parameters $m_s, q_s, q_w$. To do so, we formulate the optimization problem for the second layer weights as a Gibbs measure over the weights as in [22]:

$$\mu_\beta\left(\boldsymbol{w}\mid\{\boldsymbol{x}^\mu, y^\mu\}\right) = \frac{1}{\mathcal{Z}_\beta}e^{-\beta\left[\sum_{\mu=1}^N \ell(y^\mu, \boldsymbol{x}^\mu\cdot\boldsymbol{w})+\frac{\lambda}{2}\|\boldsymbol{w}\|_2^2\right]} = \frac{1}{\mathcal{Z}_\beta}\underbrace{\prod_{\mu=1}^N e^{-\beta\ell(y^\mu, \boldsymbol{x}^\mu\cdot\boldsymbol{w})}}_{\equiv P_y(\boldsymbol{y}|\boldsymbol{w}\cdot\boldsymbol{x}^\mu)}\underbrace{\prod_{i=1}^P e^{-\frac{\beta\lambda}{2}w_i^2}}_{\equiv P_w(\boldsymbol{w})} \tag{52}$$

Of key interest is the behavior of the free energy density, which is self-averaging in the high-dimensional limit and whose minimization gives the optimal value of the overlaps:

$$f_\beta = -\lim_{P\to\infty}\frac{1}{P}\mathbb{E}_{\{\boldsymbol{x}^\mu, y^\mu\}}\log\mathcal{Z}_\beta \tag{53}$$

To calculate the latter, we use the Replica Trick from statistical physics:

$$\mathbb{E}_{\{\boldsymbol{x}^\mu, y^\mu\}}\log\mathcal{Z}_\beta = \lim_{r\to 0}\frac{D}{Dr}\mathbb{E}_{\{\boldsymbol{x}^\mu, y^\mu\}}\mathcal{Z}_\beta^r \tag{54}$$

The right hand side can be written in terms of the *order parameters* :

$$\mathbb{E}_{\{\boldsymbol{x}^\mu, y^\mu\}}\mathcal{Z}_\beta^r = \int\frac{\mathrm{d}\rho\mathrm{d}\hat\rho}{2\pi}\int\prod_{a=1}^r\frac{\mathrm{d}m_s^a\mathrm{d}\hat{m}_s^a}{2\pi}\int\prod_{1\le a\le b\le r}\frac{\mathrm{d}q_s^{ab}\mathrm{d}\hat{q}_s^{ab}}{2\pi}\frac{\mathrm{d}q_w^{ab}\mathrm{d}\hat{q}_w^{ab}}{2\pi}e^{P\Phi^{(r)}} \tag{55}$$

$$\Phi^{(r)} = -\gamma\rho\hat\rho - \gamma\sum_{a=1}^r\sum_i m_{s,i}^a\hat{m}_{s,i}^a - \sum_{1\le a\le b\le r}\left(\gamma\sum_i q_{s,i}^{ab}\hat{q}_{s,i}^{ab} + q_w\hat{q}_w\right) \tag{56}$$

$$+ \alpha\Psi_y^{(r)}\left(\rho, m_s^a, q_s^{ab}, q_w^{ab}\right) + \Psi_w^{(r)}\left(\hat\rho, \hat{m}_s^a, \hat{q}_s^{ab}, \hat{q}_w^{ab}\right) \tag{57}$$

Where we introduced and *energetic* part $\Psi_y$ which corresponds to the likelihood of the order parameters taking a certain value based on the data, and an *entropic* part $\Psi_w$ which corresponds to the volume compatible with those order parameters :

$$\Psi_y^{(r)} = \log\int\mathrm{d}y\int\mathrm{d}\nu P_y^0(y\mid\nu)\int\prod_{a=1}^r\left[\mathrm{d}\lambda^a P_y\left(y\mid\lambda^a\right)\right]P\left(\nu, \{\lambda^a\}\right)$$
$$\Psi_w^{(r)} = \frac{1}{P}\log\int\mathrm{d}\boldsymbol{\beta}P_\beta\left(\boldsymbol{\beta}\right)e^{-\hat\rho\|\boldsymbol{\beta}\|^2}\int\prod_{a=1}^r\mathrm{d}\boldsymbol{w}^a P_w\left(\boldsymbol{w}^a\right)e^{\sum_{1\le a\le b\le r}\left[\hat{q}_w^{ab}\boldsymbol{w}^a\cdot\boldsymbol{w}^b + \sum_i\hat{q}_{s,i}^{ab}\boldsymbol{s}^a|_i\cdot\boldsymbol{s}^b|_i\right] - \sum_{a=1}^r\sum_i\hat{m}_{s,i}^a\boldsymbol{s}^a|_i\cdot\boldsymbol{\beta}|_i} \tag{58}$$

The calculation of $\Psi_y$ is exactly the same as in [22] : we defer the reader to the latter for details. As for $\Psi_w$, considering that $P_w(\boldsymbol{w}) = e^{-\frac{\lambda}{2}\|\boldsymbol{w}\|^2}$ we have (denoting the block indices as $i$):

$$\Psi_w = \lim_{P\to\infty}\mathbb{E}_{\boldsymbol{\beta},\xi,\eta}\left[\frac{1}{2}\frac{\eta^2\hat{q}_w}{\beta\lambda + \hat{V}_w} - \frac{1}{2}\log\left(\beta\lambda + \hat{V}_w\right) - \frac{1}{2P}\mathrm{tr}\log\left(\mathrm{I}_D + \hat{V}_s\Sigma\right) - \frac{1}{2P}\boldsymbol{\mu}^\top\Sigma^{-1}\boldsymbol{\mu}\right. \tag{59}$$

$$\left.+\frac{1}{2P}\left(\boldsymbol{b} + \Sigma^{-1}\boldsymbol{\mu}\right)^\top\hat{V}_s^{-1}\left(\boldsymbol{b} + \Sigma^{-1}\boldsymbol{\mu}\right) - \frac{1}{2P}\left(\boldsymbol{b} + \Sigma^{-1}\boldsymbol{\mu}\right)^\top\hat{V}_s^{-1}\left(\mathrm{I}_D + \hat{V}_s\Sigma\right)^{-1}\left(\boldsymbol{b} + \Sigma^{-1}\boldsymbol{\mu}\right)\right] \tag{60}$$

where $\hat{V}_s$ is a block diagonal matrix where the block diagonals read $(\hat{V}_{s,1}, \hat{V}_{s,2})$, and

$$\boldsymbol{b} = \left(\sqrt{\hat{q}_{s,1}}\xi_1\mathbf{1}_{\phi_1 D} + \hat{m}_{s,1}\boldsymbol{\beta}_1, \sqrt{\hat{q}_{s,2}}\xi_2\mathbf{1}_{\phi_2 D} + \hat{m}_{s,2}\boldsymbol{\beta}_2\right)$$

$$\boldsymbol{\mu} = \frac{\sqrt{\hat{q}_w}\eta}{\beta\lambda + \hat{W}_w}\frac{\mathrm{F}\mathbf{1}_P}{\sqrt{P}}$$

$$\Sigma = \frac{1}{\beta\lambda + \hat{V}_w}\frac{\mathrm{F}\mathrm{F}^\top}{P}$$

and $\lambda$ here is the coefficient in the $\ell_2$ regularization. Then, we have

$$
\begin{aligned}
\mathbb{E}_\eta \left[ \boldsymbol{\mu}^\top \boldsymbol{\Sigma}^{-1} \boldsymbol{\mu} \right] &= \frac{1}{P} \frac{\hat{q}_w}{(\beta\lambda + \hat{V}_w)^2} (\mathbf{F}\mathbf{1}_P)^\top \boldsymbol{\Sigma}^{-1} (\mathbf{F}\mathbf{1}_P) = D \frac{\hat{q}_w}{\beta\lambda + \hat{V}_w} \\
\mathbb{E}_{\eta,\xi,\boldsymbol{\beta}} \left\| \boldsymbol{b} + \boldsymbol{\Sigma}^{-1}\boldsymbol{\mu} \right\|^2 &= \sum_i \phi_i D \left( \hat{m}_{s,i}^2 + \hat{q}_{s,i} \right) + \frac{1}{P} \hat{q}_w \, \mathrm{tr}(\mathbf{F}\mathbf{F}^\top)^{-1} \\
\mathbb{E}_{\eta,\xi,\boldsymbol{\beta}} \left( \boldsymbol{b} + \boldsymbol{\Sigma}^{-1}\boldsymbol{\mu} \right)^\top \left( \mathbf{I}_D + \hat{V}_s \boldsymbol{\Sigma} \right)^{-1} \left( \boldsymbol{b} + \boldsymbol{\Sigma}^{-1}\boldsymbol{\mu} \right) &= \frac{1}{P} \hat{q}_w \, \mathrm{tr} \left[ \mathbf{F}\mathbf{F}^\top \left( \mathbf{I}_D + \hat{V}_s \boldsymbol{\Sigma} \right)^{-1} \right] \\
&\quad + \sum_i \left( \hat{m}_{s,i}^2 + \hat{q}_{s,i} \right) \mathrm{tr} \left( \mathbf{I}_D + \hat{V}_{s_i} \boldsymbol{\Sigma}_i \right)^{-1}
\end{aligned}
\tag{61}
$$

$$
\Psi_w = -\frac{1}{2} \log \left( \beta\lambda + \hat{V}_w \right) - \frac{1}{2} \lim_{P\to\infty} \frac{1}{P} \, \mathrm{tr} \log \left( \mathbf{I}_D + \frac{\hat{V}_s}{\beta\lambda + \hat{V}_w} \frac{\mathbf{F}\mathbf{F}^\top}{P} \right)
\tag{62}
$$

$$
+ \sum_i \frac{\hat{m}_{s,i}^2 + \hat{q}_{s,i}}{2\hat{V}_{s,i}} [\phi_i \gamma] - \lim_{P\to\infty} \frac{1}{P} \, \mathrm{tr} \left( (\hat{m}_s^2 + \hat{q}_s) \left( \mathbf{I}_D + \frac{\hat{V}_s}{\beta\lambda + \hat{V}_w} \frac{\mathbf{F}\mathbf{F}^\top}{P} \right)^{-1} \right)
\tag{63}
$$

$$
+ \frac{1}{2} \frac{\hat{q}_w}{\beta\lambda + \hat{V}_w} [1 - \gamma] + \frac{\hat{q}_w}{2} \lim_{P\to\infty} \frac{1}{P} \left[ \mathrm{tr} \left( \hat{V}_s^{-1}(\mathbf{F}\mathbf{F}^\top)^{-1} \right) - \mathrm{tr} \left( (\hat{V}_s \mathbf{F}\mathbf{F}^\top)^{-1} \left( \mathbf{I}_D + \frac{\hat{V}_s}{\beta\lambda + \hat{V}_w} \frac{\mathbf{F}\mathbf{F}^\top}{P} \right)^{-1} \right) \right]
\tag{64}
$$

where $(\hat{m}_s^2 + \hat{q}_s)$ and $\hat{V}_s$ in the trace operator are to be understood as diagonal matrices here. Using the following simplification

$$
\mathrm{tr} \left( V_s^{-1}(\mathbf{F}\mathbf{F}^\top)^{-1} \right) - \mathrm{tr} \left( (V_s \mathbf{F}\mathbf{F}^\top)^{-1} \left( \mathbf{I}_D + \frac{\hat{V}_s}{\beta\lambda + \hat{V}_w} \frac{\mathbf{F}\mathbf{F}^\top}{P} \right)^{-1} \right)
\tag{65}
$$

$$
= \frac{1}{P(\beta\lambda + \hat{V}_w)} \, \mathrm{tr} \left( \mathbf{I}_d + \frac{\hat{V}_s}{\beta\lambda + \hat{V}_w} \frac{\mathbf{F}\mathbf{F}^\top}{P} \right),
\tag{66}
$$

we finally obtain

$$
\Psi_w = -\frac{1}{2} \log \left( \beta\lambda + \hat{V}_w \right) - \frac{1}{2} \lim_{P\to\infty} \frac{1}{P} \, \mathrm{tr} \log \left( \mathbf{I}_D + \frac{\hat{V}_s}{\beta\lambda + \hat{V}_w} \frac{\mathbf{F}\mathbf{F}^\top}{P} \right)
\tag{67}
$$

$$
+ \sum_i \frac{\hat{m}_{s,i}^2 + \hat{q}_{s,i}}{2\hat{V}_{s,i}} [\phi_i \gamma] - \lim_{P\to\infty} \frac{1}{P} \, \mathrm{tr} \left( (\hat{m}_s^2 + \hat{q}_s) \left( \mathbf{I}_D + \frac{\hat{V}_s}{\beta\lambda + \hat{V}_w} \frac{\mathbf{F}\mathbf{F}^\top}{P} \right)^{-1} \right)
\tag{68}
$$

$$
+ \frac{1}{2} \frac{\hat{q}_w}{\beta\lambda + \hat{V}_w} \left[ 1 - \gamma + \lim_{P\to\infty} \frac{1}{P} \, \mathrm{tr} \left( \left( \mathbf{I}_D + \frac{\hat{V}_s}{\beta\lambda + \hat{V}_w} \frac{\mathbf{F}\mathbf{F}^\top}{P} \right)^{-1} \right) \right]
\tag{69}
$$

.

Define $M = \frac{1}{P} \hat{V}_s F F^\top$. $\Psi_w$ involves the following term :

$$
\frac{1}{P} \, \mathrm{tr} \left( \mathbf{I}_D + \frac{\hat{V}_s}{\beta\lambda + \hat{V}_w} M \right)^{-1} = \gamma(\beta\lambda + \hat{V}_w) g(-\beta\lambda + \hat{V}_w)
\tag{70}
$$

Where $g$ is the Stieljes transform of $M$ : $g(z) = \frac{1}{D} \mathrm{Tr}(z - M)^{-1}$.

## B.4   Some random matix theory for block matrices

To calculate the desired Stieljes transform, we specialize to the case of Gaussian random feature matrices and use again tools from Statistical Physics.

**Isotropic setup**    We first consider the isotropic setup where $\hat{V}_s$ is a scalar. Then we have

$$g(z) = \frac{1}{D} \operatorname{Tr}(z - M)^{-1} \tag{71}$$

$$= -\frac{1}{D} \frac{d}{dz} \operatorname{Tr} \log(z - M) \tag{72}$$

$$= -\frac{1}{D} \frac{d}{dz} \log \det(z - M) \tag{73}$$

$$= -\frac{2}{D} \frac{d}{dz} \left\langle \log \int dy \, e^{-\frac{1}{2} y(z-M)y^\top} \right\rangle \tag{74}$$

where $\langle . \rangle$ stands for the average over disorder, here the matrix $F$. Then we use the replica trick,

$$\langle \log Z \rangle \to_{n\to 0} \frac{1}{n} \log \langle Z^n \rangle \tag{75}$$

Therefore we need to calculate $Z^n$:

$$\langle Z^n \rangle = \int \prod_{a=1}^{n} d\vec{y}_a e^{-\frac{1}{2} z \sum_a \vec{y}_a^2} \int dF e^{-\frac{1}{2} \mathrm{FLF}^\top} \tag{76}$$

$$= \int \prod_{a=1}^{n} d\vec{y}_a e^{-\frac{1}{2} z \sum_a \vec{y}_a^2} (\det L)^{-P/2} \tag{77}$$

where $L = \mathbb{I}_d - \frac{\gamma \hat{V}_s}{D} \sum_a y_a \vec{y}_a^\top$. Here we decompose the vectors $\vec{y}$ into two parts. Then we use that $\det L = \det \tilde{L}$, where $\tilde{L}_{ab} = \delta_{ab} - \frac{\gamma \hat{V}_s}{D} \vec{y}_a \cdot \vec{y}_b$. Then,

$$\langle Z^n \rangle = \int \prod_a d\vec{y}_a e^{\frac{1}{2} \sum_a \vec{y}_a \cdot \vec{y}_a} \det \left[ \mathbb{I}_d - \frac{\gamma \hat{V}_s}{D} Y^\top Y \right]^{-\frac{P}{2}} \tag{78}$$

where $Y \in \mathbb{R}^{d \times n}$ with $Y_{ia} = y_i^a$. We introduce $1 = \int dQ_{ab} \delta(dQ_{ab} - \vec{y}_a \cdot \vec{y}_b)$, and use the Fourier representation of the delta function, yielding

$$\langle Z^n \rangle = \int dQ e^{-\frac{D}{2} z \operatorname{Tr} Q} \left( \det \left[ \mathbb{I} - \hat{V}_s Q \right] \right)^{-\frac{P}{2}} \int d\hat{Q}_{ab} e^{\sum_{ab} dQ_{ab} \hat{Q}_{ab} - \hat{Q}_{ab} \vec{y}_a \vec{y}_b} \tag{79}$$

$$= \int dQ d\hat{Q} e^{-dS[Q,\hat{Q}]} \tag{80}$$

where

$$S[Q, \hat{Q}] = \frac{1}{2} z \operatorname{Tr} Q + \frac{1}{2\gamma} \log \det \left( 1 - \hat{V}_s Q \right) + \frac{1}{2} \log \det \left( 2\hat{Q} \right) - \operatorname{Tr} \left( Q\hat{Q} \right) \tag{81}$$

A saddle-point on $\hat{Q}$ gives $Q = (2\hat{Q})^{-1}$. Therefore we can replace in $S$,

$$S[Q] = \frac{1}{2} z \operatorname{Tr} Q + \frac{1}{2\gamma} \log \det \left( 1 - \hat{V}_s Q \right) - \frac{1}{2} \log \det (Q) \tag{82}$$

In the RS ansatz $Q_{ab} = q\delta_{ab}$, this yields

$$S[q]/n = \frac{1}{2} zq + \frac{1}{2\gamma} \log \left( 1 - \hat{V}_s q \right) - \frac{1}{2} \log q \tag{83}$$

Now we may apply a saddle point method to write

$$\langle Z^n \rangle = e^{-dS[q^\star]} \tag{84}$$

where $q^\star$ minimizes the action, i.e. $\frac{dS}{dq}\big|_{q^\star} = 0$:

$$z - \frac{\hat{V}_s}{1 - \hat{V}_s q^\star} - \frac{1}{q^\star} = 0 \tag{85}$$

Therefore,

$$g(z) = -\frac{2}{D} \frac{d}{dz} (-DS[q^\star]) = q^\star(z) \tag{86}$$

**Anisotropic setup**   Now we consider that $V$ is a black diagonal matrix, with blocks of size $\phi_i d$ with values $\hat{V}_{s,i}$. We need to adapt the calculation by decomposing the auxiliary fields $y$ along the different blocks, then define separately the overlaps of the blocks $q_i$. Then we obtain the following action:

$$S[\{q_i\}]/n = \frac{1}{2}z\sum_i \phi_i q_i + \frac{1}{2\gamma}\log\left(1 - \sum_i \phi_i \hat{V}_{s,i}q_i\right) - \sum_i \frac{\phi_i}{2}\log(q_i) \tag{87}$$

To obtain the Stieljes transform, we need to solve a system of coupled equations :

$$g(z) = \sum_i \phi_i q_i^\star \tag{88}$$

$$\phi_i z\Omega q_i^\star - \phi_i \hat{V}_i q_i^\star - \phi_i \Omega = 0 \tag{89}$$

$$\Omega = 1 - \sum_i \phi_i \hat{V}_{s,i}q_i^\star \tag{90}$$

We therefore conclude that

$$\lim_{P\to\infty}\frac{1}{P}\,\mathrm{tr}\left(\left(\mathrm{I}_D + \frac{\hat{V}_s}{\beta\lambda+\hat{V}_w}\frac{\mathrm{FF}^\top}{P}\right)^{-1}\right) = \gamma(\beta\lambda+\hat{V}_w)\sum_i \phi_i q_i^\star \tag{91}$$

$$\lim_{P\to\infty}\frac{1}{P}\,\mathrm{tr}\left((\hat{m}_s^2+\hat{q}_s)\left(\mathrm{I}_D + \frac{\hat{V}_s}{\beta\lambda+\hat{V}_w}\frac{\mathrm{FF}^\top}{P}\right)^{-1}\right) = \gamma(\beta\lambda+\hat{V}_w)\sum_i (\hat{m}_{s,i}^2+\hat{q}_{s,i})\phi_i q_i^\star \tag{92}$$

## B.5   Obtaining the saddle-point equations

The saddle-point equations of [22] become the following in the anisotropic setup :

$$\begin{cases}
\hat{r}_{s,i} = -2\sigma_{x,i}\frac{\alpha}{\gamma}\partial_{r_{s,i}}\Psi_y(R,Q,M) & r_{s,i} = -\frac{2}{\gamma}\partial_{\hat{r}_{s,i}}\Psi_w\left(\hat{r}_{s,i},\hat{q}_{s,i},\hat{m}_{s,i},\hat{r}_w,\hat{q}_w\right)\\
\hat{q}_{s,i} = -2\sigma_{x,i}\frac{\alpha}{\gamma}\partial_{q_{s,i}}\Psi_y(R,Q,M) & q_{s,i} = -\frac{2}{\gamma}\partial_{\hat{q}_{s,i}}\Psi_w\left(\hat{r}_{s,i},\hat{q}_{s,i},\hat{m}_{s,i},\hat{r}_w,\hat{q}_w\right)\\
\hat{m}_{s,i} = \sigma_{x,i}\frac{\alpha}{\gamma}\partial_{m_{s,i}}\Psi_y(R,Q,M) & m_{s,i} = \frac{1}{\gamma}\partial_{\hat{m}_{s,i}}\Psi_w\left(\hat{r}_{s,i},\hat{q}_{s,i},\hat{m}_{s,i},\hat{r}_w,\hat{q}_w\right)\\
\hat{r}_w = -2\alpha\partial_{r_w}\Psi_y(R,Q,M) & r_w = -2\partial_{\hat{r}_w}\Psi_w\left(\hat{r}_{s,i},\hat{q}_{s,i},\hat{m}_{s,i},\hat{r}_w,\hat{q}_w\right)\\
\hat{q}_w = -2\alpha\partial_{q_w}\Psi_y(R,Q,M) & q_w = -2\partial_{\hat{q}_w}\Psi_w\left(\hat{r}_{s,i},\hat{q}_{s,i},\hat{m}_{s,i},\hat{r}_w,\hat{q}_w\right)
\end{cases} \tag{93}$$

The saddle point equations corresponding to $\Psi_w$ do not depend on the learning task, and can be simplified in full generality to the following set of equations :

$$\begin{cases}
V_{s,i} = \frac{1}{\hat{V}_{s,i}}\left(\phi_i - z_i g_\mu(-z_i)\right)\\
q_{s,i} = \frac{\sigma_{\beta,i}\hat{m}_{s,i}^2+\hat{q}_{s,i}}{\hat{V}_{s,i}^2}\left[\phi_i - 2z_i g_\mu(-z_i) + z_i^2 g_\mu'(-z_i)\right] - \frac{\hat{q}_w}{(\beta\lambda+\hat{V}_w)\hat{V}_{s,i}}\left[-z_i g_\mu(-z_i) + z_i^2 g_\mu'(-z_i)\right]\\
m_{s,i} = \frac{\sigma_{\beta,i}\hat{m}_{s,i}}{\hat{V}_{s,i}}\left(\phi_i - z_i g_\mu(-z_i)\right)\\
V_w = \sum_i \frac{\gamma}{\beta\lambda+\hat{V}_w}\left[\frac{1}{\gamma} - 1 + z_i g_\mu(-z_i)\right]\\
q_w = \sum_i \frac{\sigma_{\beta,i}\hat{m}_s^2+\hat{q}_{s,i}}{(\beta\lambda+\hat{V}_w)\hat{V}_{s,i}}\left[-z_i g_\mu(-z_i) + z_i^2 g_\mu'(-z_i)\right] + \gamma\frac{\hat{q}_w}{(\beta\lambda+\hat{V}_w)^2}\left[\frac{1}{\gamma} - 1 + z_i^2 g_\mu'(-z_i)\right]
\end{cases} \tag{94}$$

As for the saddle point equations corresponding to $\Psi_y$, they depend on the learning task. We can solve analytically for the two setups below.

The solution of the saddle-point equations allow to obtain the order parameter values that determine the covariance of $\nu$ and $\lambda$ (for one given replica, say $a = 1$) and hence to compute the test error as explained at the beginning of the appendix.

**Square loss regression**   Let us first specialize to a simple regression setup where the teacher is a Gaussian additive channel :

$$\ell(y,x) = \frac{1}{2}(x - y)^2 \tag{95}$$

$$\mathcal{P}(x|y) = \frac{1}{\sqrt{2\pi\Delta}}e^{-\frac{(x-y)^2}{2\Delta}} \tag{96}$$

In this case, the saddle-point equations simplify to :

$$\begin{cases} \hat{V}_{s,i}^\infty = \sigma_{x,i}\frac{\alpha}{\gamma}\frac{\kappa_{1,i}^2}{1+V^\infty} \\ \hat{q}_{s,i}^0 = \sigma_{x,i}\alpha\kappa_{1,i}^2\frac{\phi_i\sigma_{\beta,i}+\Delta+Q^\infty-2M^\infty}{\gamma(1+V^\infty)^2} \\ \hat{m}_{s,i} = \sigma_{x,i}\frac{\alpha}{\gamma}\frac{\kappa_{1,i}}{1+V^\infty} \\ \hat{V}_w^\infty = \frac{\alpha\sum_i\phi_i\kappa_{\star,i}^2}{1+V^\infty} \\ \hat{q}_w^\infty = \alpha\sum_i\phi_i\kappa_{\star,i}^2\frac{1+\Delta+Q^\infty-2M^\infty}{(1+V^\infty)^2} \end{cases} \tag{97}$$

**Square loss classification**   Next we examine the classification setup where the teacher gives binary labels with a sign flip probability of $\Delta$, and the student learns them through the mean-square loss:

$$\mathcal{P}(x|y) = (1-\Delta)\delta(x-\text{sign}(y)) + \Delta\delta(x+\text{sign}(y)), \quad \Delta \in [0,1] \tag{98}$$

$$\ell(y,x) = \frac{1}{2}(x-y)^2 \tag{99}$$

In this case, the equations simplify to :

$$\begin{cases} \hat{V}_{s,i}^\infty = \sigma_{x,i}\frac{\alpha}{\gamma}\frac{\kappa_{1,i}^2}{1+V^\infty} \\ \hat{q}_{s,i}^0 = \sigma_{x,i}\alpha\kappa_{1,i}^2\frac{\phi_i\sigma_{\beta,i}+Q^\infty-2\frac{(1-2\Delta)\sqrt{2}}{\sqrt{\pi}}M^\infty}{\gamma(1+V^\infty)^2} \\ \hat{m}_{s,i} = \sigma_{x,i}\frac{\alpha}{\gamma}\frac{\kappa_{1,i}}{1+V^\infty} \\ \hat{V}_w^\infty = \frac{\alpha\sum_i\phi_i\kappa_{\star,i}^2}{1+V^\infty} \\ \hat{q}_w^\infty = \alpha\sum_i\phi_i\kappa_{\star,i}^2\frac{1+Q^\infty-2\frac{(1-2\Delta)\sqrt{2}}{\sqrt{\pi}}M^\infty}{(1+V^\infty)^2} \end{cases} \tag{100}$$

**Logistic loss classification**   For general loss functions such as the cross-entropy loss, $\ell(x,y) = \log(1 + e^{-xy})$, the saddle-point equations for $\Psi_y$ do not simplify and one needs to evaluate the integrals over $\xi$ numerically.

## B.6   Training loss

To calculate the training loss, we remove the regularization term:

$$\epsilon_t = \frac{1}{N}\mathbb{E}_{\{\boldsymbol{x}^\mu,y^\mu\}}\left[\sum_{\mu=1}^N \ell\left(y^\mu, \boldsymbol{x}^\mu\cdot\hat{\boldsymbol{w}}\right)\right] \tag{101}$$

As explained in [22], the latter can be written as

$$\epsilon_t = \mathbb{E}_\xi\left[\int_{\mathbb{R}}\mathrm{d}y\,\mathcal{Z}_y^0\left(y,\omega_0,V_0\right)\ell\left(y,\eta\left(y,\omega_1\right)\right)\right], \quad \xi\sim\mathcal{N}(0,1) \tag{102}$$

where $\mathcal{P}$ is the teacher channel defined in (12), and we have:

$$\omega_0 = M/\sqrt{Q}\xi \tag{103}$$

$$V_0 = \rho - M^2/Q \tag{104}$$

$$\omega_1 = \sqrt{Q}\xi \tag{105}$$

$$\eta(y,\omega) = \arg\min_x \frac{(x-\omega)^2}{2V} + \ell(y,x) \tag{106}$$

$$\mathcal{Z}_y^0(y,\omega) = \int\frac{\mathrm{d}x}{\sqrt{2\pi V_0}}\,e^{-\frac{1}{2V_0}(x-\omega)^2}\mathcal{P}(y|x) \tag{107}$$

**Square loss regression**   Consider again the regression setup where the teacher is a Gaussian additive channel $\mathcal{P}(x|y) = \frac{1}{\sqrt{2\pi\Delta}}e^{-\frac{(x-y)^2}{2\Delta}}$ and the loss is $\ell(y,x) = \frac{1}{2}(x-y)^2$. These assumptions imply

$$\mathcal{Z}_y^0(y,\omega) = \frac{1}{\sqrt{2\pi(V_0+\Delta)}}e^{-\frac{(y-\omega)^2}{2(V_0+\Delta)}}, \tag{108}$$

$$\eta(y,\omega) = \frac{\omega+Vy}{1+V}, \tag{109}$$

which yields the simple formula for the training error

$$\epsilon_t = \frac{1}{2(1+V)^2} \mathbb{E}_\xi \left[ \int dy \mathcal{Z}_y^0 (y, \omega) (y - \omega_1)^2 \right] \tag{110}$$

$$= \frac{1}{2(1+V)^2} \mathbb{E}_\xi \left[ V_0 + \Delta + (\omega_0 - \omega_1)^2 \right] \tag{111}$$

$$= \frac{1}{2(1+V)^2} (\rho + Q - 2M + \Delta) \tag{112}$$

$$= \frac{\epsilon_g + \Delta}{(1+V)^2}. \tag{113}$$

Notice that the training loss is closely related to the test loss; inverting the latter expression, we have $\epsilon_g = (1 + V^2)\epsilon_t - \Delta$, showing that $V$ acts as a variance term opening up a generalization gap.

**Square loss classification**  Next we specialize to our classification case study where the teacher is a Gaussian additive channel $\mathcal{P}(x|y) = (1 - \Delta)\delta(x - \text{sign}(y)) + \Delta\delta(x + \text{sign}(y))$ and the loss is $\ell(y, x) = \frac{1}{2}(x - y)^2$. These assumptions imply

$$\mathcal{Z}_y^0 (y, \omega) = \frac{1}{\sqrt{2\pi V_0}} \left( (1 - \Delta)e^{-\frac{(y-\omega)^2}{2V_0}} + \Delta e^{-\frac{(-y-\omega)^2}{2V_0}} \right), \tag{114}$$

$$\eta(y, \omega) = \frac{\omega + Vy}{1 + V}, \tag{115}$$

which yields the simple formula for the training error

$$\epsilon_t = \frac{1}{2(1+V)^2} \mathbb{E}_\xi \left[ V_0 + (1 - \Delta)(\omega_0 - \omega_1)^2 + \Delta(\omega_0 + \omega_1)^2 \right] \tag{116}$$

$$= \frac{1}{2(1+V)^2} (\rho + Q - 2(1 - 2\Delta)M) \tag{117}$$

$$\tag{118}$$

This expression is similar to the one obtained in the regression setup, except for the role of the noise, which reflects label flipping instead of additive noise. As a sanity check, note that flipping all the labels, i.e. $\Delta = 1$, is equivalent to the transformation $M \to -M$, as one could expect.