# OpenReview forum: "On the interplay between data structure and loss function in classification problems"
_NeurIPS.cc/2021/Conference — NeurIPS 2021 Poster_

### Official Review · Reviewer_Xc21 · 2021-07-06

**Rating:** 6
**Confidence:** 4

**Summary:**

The authors study the loss achieved by random feature models that learns from simple (but structured) data.
The main result consists in the comparison of two losses (square and logistic) under three settings in the data structure (misaligned, isotropic and aligned that roughly represent the difficulty of the task).
The paper is well written and presents a thorough analysis of the related works.

**Limitations And Societal Impact:**

It is a simple yet instructive model.
No societal impact is foreseen.

**Main Review:**

The paper uses standard and new tools of statistical physics to address the problem summarized above.

The comparison between the two losses is very interesting. A direct comparison between the results would be unfair and the authors make nice considerations to convince of the benefits of logistic loss. Considerations that are confirmed in the experimental setup on real data.

In the end, the authors conclude that their results may be limited to the random feature models as there is a contrast with the results of ref.[51] about DNNs. I do not understand why the authors did not include this analysis in their paper using the framework of ref.[32]. I strongly believe that it would be more natural to answer these questions in this paper instead of waiting for future works.
add
In the end, the role of the data structure seems to play a minor role in the paper (and in the results). Apart from a shift of the curves and a counter-intuitive result on the regularization effect of misalignment in the square loss, I do not see many considerations on this aspect.
Is it because the model is still too simple? The author should extend their discussion and maybe analysis to highlight the importance of data structure.

My impression is that, although it is pointing in the right direction, the paper needs more results to meet the acceptance bar.

################# after rebuttal #################

Thanks to the authors for the clarification and the promised changes.

The authors are indeed right about ref.32, the analysis I suggest is more complex than what appeared in the first place. Nevertheless, it may be worth integrating their analytical result with some empirical ones and give a partial answer to the questions they asked. I still believe it is more natural to provide an answer (although partial) in this paper, instead of waiting for future works.


**Time Spent Reviewing:**

4

---

> ### Author Response · Authors · 2021-08-09
> **Rebuttal**
>
>
> We thank the reviewer for their valuable feedback.
>
> While we agree that it would be very interesting to bridge the gap with the realistic setup of ref. 51, it remains a strong theoretical hurdle. Perhaps our citation of ref. 32 lead to a misunderstanding; while this paper studies general features maps, these feature maps are fixed at the time of analysis and it is not currently possible to analyse multi-layer training with the replica framework. While we could have also chosen to consider classification from these learned feature maps, they require a stochastic approximation step and therefore do not provide a fully analytic result as in our work. Even then, the leap would still be rather large with ref. 51, as the latter studied state-of-the-art multi-class tasks; additionally, to make the square loss tractable, the authors of ref. 51 had to rescale the importance of the incorrect classes, which is likely to play an important role. We revised the conclusion to reflect these points and add clarity.
>
> Regarding the impact of data structure, our goal is indeed to highlight it as much as possible. In fact, we show that the logistic loss is best suited to take advantage of the low dimensionality of a task, when the data structure of inputs and the structure of the target rule are aligned. This is very well visualized in the phase spaces of Appendix A. To emphasize this point more strongly, we will move parts of Appendix A to the main text (as also discussed with reviewer 2).

---

### Official Review · Reviewer_LCY7 · 2021-07-11

**Rating:** 5
**Confidence:** 4

**Summary:**

This paper used the replica method to derive the high dimensional asymptotics of the test error of random features model in the strong and weak features model with square loss and logistic loss. The observations drawn from plotting the analytical prediction are the following: 1. When the parameters vector of the target function is aligned with the covariance structure of the covariates, the generalization error is smaller (given fixed N/D, P/D, lambda, loss function). 2. In the aligned settings, the test error of using the logistic loss is smaller than that of using the square loss (in a specific setting). This paper also performed experiments on parity MNIST and CIFAR10(2), and the experimental result coincides with the results with idealized assumptions.

**Limitations And Societal Impact:**

In general, the reviewer suggests the authors consider more scenarios and perform a more thorough investigation to support their conclusions.

**Main Review:**

Overall, this is a decent result. The theoretical part which is the replica calculation seems like a straightforward generalization of previous results (it didn’t appear in the literature though). The strong and weak features model setup in the proportional limit has not been fully investigated before, so drawing some useful conclusions from the analytical prediction would be very interesting.

An issue with this paper is that it drew conclusions based on a very specific choice of parameters and specific settings. It is not clear whether this is representative enough to lead to the conclusions.
1) In the figures, this paper just chooses N = D. It is not clear whether the conclusion still holds for some larger N/D (in practice, the sample size is several times larger than the dimension). In the appendix, there seem to be more experiments for other N/D, but the author didn’t use these experimental results to support the conclusions drawn in this paper. The reviewer suggests the author draw necessary figures for other N/D to demonstrate that the conclusions consistently hold.
2) This paper just fixes lambda to be a small parameter like 10^{-3} or 10^{-4}. However, it is not clear whether one should compare logistic/square loss by choosing a vanishing regularization. Another scenario would be to choose the optimal lambda that minimizes the test error and then compare the logistic/square loss.

Other issues:
1. In Eq (8), please explain what is x_i. Is it the eigenvectors of top PCs, or the coordinates of the covariates?

**Time Spent Reviewing:**

2.5 hours

---

> ### Author Response · Authors · 2021-08-09
> **Rebuttal**
>
> We thank the reviewer for their valuable feedback.
>
> We acknowledge the need of a wide range of experimental settings to support claims. However, even such a simplistic model as strong and weak features has a lot of free parameters, and investigating each one of them is difficult given the space constraints. To avoid excessive cluttering, we focused on varying the parameters controlling the structure of data (anisotropy parameters and label noise) in the main text, and extensively studied the impact of P/D and N/D in appendix A. However, we recognize the need for a discussion of how those other parameters affect our conclusions in the main text.
>
> The better performance of logistic loss as compared to the square loss for low-dimensional tasks can be confirmed by examining the phase spaces of Fig. 7 and 8:
> For square loss, even in the noiseless scenario (7c), overfitting appears in the overparametrized region (P>N).
> For logistic loss, the implicit regularization described in the main text fully cancels this overfitting.
> The difference between logistic and square loss we discuss in the paper holds everywhere in the overparametrized regime, but most visibly for a large band of N/D ranging from 0.01 to 100. We agree with the reviewer that it is important to highlight this result, and will move parts of appendix A to the main text.
>
> As for the role of the regularization constant, we agree that it is not clear how to provide a fair comparison between logistic loss and square loss. We chose to focus on the near-vanishing regularization regime in the main text, but agree that it would be interesting to compare them at optimal regularization; we will add a section on this in the revised manuscript.
>
> [1] d’Ascoli et al. 2020, Triple descent and the two kinds of overfitting: where and why do they appear?

---

### Official Review · Reviewer_WQoW · 2021-07-19

**Rating:** 7
**Confidence:** 3

**Summary:**

This paper studies the generalization in an analytically solvable setup: a two-layer network with random features on the first layer and learnable features on the second layer, under squared loss and logistic loss. The authors derive asymptotic expressions for the train and test errors of the above model trained on a dataset with block covariance matrix with tunable saliency (for input) and noise (for mapping). In experiments, the authors studied and compared how the data structure and loss function affect the double descent, generalization, and also test it in the case of MNIST and CIFAR10 datasets.

**Limitations And Societal Impact:**

The authors address the limitation that their analytical results are based on non-rigorous tools from statistical physics. It would be better if the authors can also discuss how their results and insights can generalize over more complex models and datasets.

**Main Review:**

On the theoretical side, I did not check the proof of the theorems, but I believe that the derivation is correct and reasonable. I think the theoretical contribution is novel. For clarity, it would make the paper much more insightful if the authors can provide clear intuition about the cause of the double descent using the main analytical result, and why in noiseless scenario with logistic loss this phenomenon disappear.

On the experiment side, I think the experiment is well-conducted.

The authors note that their results appear to be in stark conflict with those of (Hui and Belkin, 2020), which train on large-scale models. Could the authors address the following question: to which extend can the results in this two-layer network generalize to larger models and more complex datasets?

Overall, I think the paper is well-written, novel, and makes a step that offers interesting technique to address the generalization of NNs on structured datasets.

**Time Spent Reviewing:**

3 hours

---

> ### Author Response · Authors · 2021-08-09
> **Rebuttal**
>
> We thank the reviewer for their positive feedback and useful comments.
>
> As discussed in Sec. 4, intuition on why double descent can be gained by inspecting the order parameters of Fig. 5. For square and logistic loss, the peak of double descent is related to the increase in the norm of the estimator, Q. In both cases, in absence of regularisation, Q diverges when P reaches the interpolation threshold because of overfitting. This phenomenon, that we find analytically, has been observed in simpler models, see Refs [5,41] and discussed heuristically for hinge-loss in [13].
> For the logistic loss, the peak is masked by the fact that Q increases continuously as we overparametrize, due to the overconfidence. Noise enhances the overfitting at the interpolation threshold, and this makes the peak emerge even for the logistic loss in that case.  We agree that this is an important point and we will extend the discussion in the revised version.
>
> Regarding the connection to more complex models and datasets, we believe that a first step to bridge the gap would be to address the case of two-layer networks where both layers are trained. Unfortunately, a theoretical analysis of these two-layer networks remains currently challenging for most frameworks of analysis including ours. We became aware of  the work of [1]  that shows that feature learning helps models capture the low-dimensional structures of the inputs, potentially compensating the flaws of the quadratic loss in this regard. We will extend this discussion in the revised manuscript.
>
> [1] Aristide Baratin et al., Implicit regularization in deep learning:  A view from function space

---

### Decision · Program_Chairs · 2021-09-27

**Decision:**

Accept (Poster)

**Comment:**

The paper presents an interesting analytical result on generalization which all three reviewers considered worthwhile.
Based on the given scores (5,6,7) and the fact that the machine learning field would likely benefit from some more theoretical contributions I recommend acceptance as a poster presentation.